# Learning from data to design functional materials without inversion symmetry

Prasanna V. Balachandran[1], Joshua Young[2], Turab Lookman[1] & James M. Rondinelli[3]

Accelerating the search for functional materials is a challenging problem. Here we develop an informatics-guided *ab initio* approach to accelerate the design and discovery of noncentrosymmetric materials. The workflow integrates group theory, informatics and density-functional theory to uncover design guidelines for predicting noncentrosymmetric compounds, which we apply to layered Ruddlesden-Popper oxides. Group theory identifies how configurations of oxygen octahedral rotation patterns, ordered cation arrangements and their interplay break inversion symmetry, while informatics tools learn from available data to select candidate compositions that fulfil the group-theoretical postulates. Our key outcome is the identification of 242 compositions after screening $\sim 3,200$ that show potential for noncentrosymmetric structures, a 25-fold increase in the projected number of known noncentrosymmetric Ruddlesden-Popper oxides. We validate our predictions for 19 compounds using phonon calculations, among which 17 have noncentrosymmetric ground states including two potential multiferroics. Our approach enables rational design of materials with targeted crystal symmetries and functionalities.

[1] Theoretical Division, Los Alamos National Laboratory, Los Alamos, New Mexico 87545, USA. [2] Department of Materials Science and Engineering, Drexel University, Philadelphia, Pennsylvania 19104, USA. [3] Department of Materials Science and Engineering, Northwestern University, Evanston, Illinois 60208, USA. Correspondence and requests for materials should be addressed to P.V.B. (email: pbalachandran@lanl.gov) or to J.M.R. (email: jrondinelli@northwestern.edu).

Noncentrosymmetric (NCS) oxide ceramics that break all improper rotations and centres of symmetry are challenging to discover. Materials with polar, piezoelectric, chiral and those exhibiting circular dichroism (collectively referred to as NCS materials) are defined by the absence of inversion symmetry and are present everywhere—in the form of organic amino acids, sugars and other biological molecules[1]. Inorganic NCS materials containing oxide anions are also not uncommon[2]. Quartz crystals with a helical arrangement of corner-connected $SiO_4$ tetrahedral units maintain the punctuality of our mechanical timepieces[3]. At inorganic crystalline surfaces, chirality plays a crucial role in corrosion processes, heterogeneous catalysis and the fidelity of enantioselective-based production or separation of industrial solvents, plastics and pharmaceutical drugs[4]. $Pb(Zr,Ti)O_3$, $BaTiO_3$ and $BiFeO_3$ are some of the archetypal polar oxides that have impacted many critical technologies[5]. Often inorganic polar and chiral basic building units (BBUs) are selected and assembled together, but acentric organization of BBUs within a unit cell are difficult to predict due to the complex interplay of chemistry and structure.

In the context of inorganic oxides, which is the focus of this work, the design of NCS materials has relied mainly on BBUs with metal centres that have $d^0$ electronic configurations or lone-pair cations, where the acentricity arises from an electronic origin due to the pseudo- or second-order Jahn–Teller (SOJT) effect[6,7]. A majority of inorganic oxides, however, strongly prefer close-packed arrangements of ions and highly symmetric cation coordination environments (for example, octahedra). This is mainly due to the dominant electrostatic effects that are optimized by favouring like–unlike interactions (that is, positive and negative dipoles align equally and oppositely), which stabilize atomic arrangements with inversion symmetry[8]. In fact, the presence of BBUs with $d^0$ metal centres alone is not a sufficient condition for designing NCS materials. For example, the perovskite $SrTiO_3$ is a quantum paraelectric or incipient ferroelectric[9], whereas the isoelectronic layered Ruddlesden-Popper (RP) $Sr_2TiO_4$ is a centric dielectric[10]. Hence, it is the complex interplay between structure and chemistry that determines the formation of NCS inorganic oxides.

Alternative to pseudo-JT or SOJT effects, the 'trilinear coupling', 'hybrid improper' or 'geometric ferroelectricity' mechanism, where two nonpolar lattice distortions (octahedral rotations or tilting) couple to a polar lattice mode, have also been shown to break the inversion symmetry with interesting technological consequences[11,12]. Even in this case, no a priori rules exist that guide the design of new hybrid improper ferroelectric materials, unless exhaustive calculations are carried out to map the chemical and energy landscape that subsequently inform experiments[12]. As a result, NCS inorganic oxides are challenging to discover.

Although high-throughput first principles-based methods have shown promise in the design of NCS half-Heusler alloys[13], exhaustive calculations for more complex crystal structures with numerous polymorphs (such as the RPs) and thousands of unexplored chemical compositions have not (yet) been demonstrated. This is partly because the potential energy surface of complex oxides is difficult to navigate. Phonon instabilities at high-symmetry points away from the Γ-point in the irreducible Brillouin zones cause the primitive unit cell to multiply several fold, resulting in large system sizes and vast numbers of unique atomic arrangements. It is challenging to rigorously evaluate the energetics of all structures in a high-throughput manner. Furthermore, chemistries with partially filled $d$ (and/or $f$) orbitals and the existence of energetically competing ground states complicate the structure prediction process. As a result, novel approaches are desired to guide the first principles calculations in an effective manner. Materials informatics, a growing field at the intersections of many scientific disciplines including data and information science, statistics, machine learning (ML) and optimization, has the potential to accomplish this objective[14].

Here we develop a predictive data-driven computational framework that unites applied group theory, informatics techniques and ab initio electronic structure calculations for designing novel NCS materials. We apply it to the two-dimensional $n=1$ RP structure family (Fig. 1a), for which to date few compositions exist in NCS crystal classes[15–17]. Nonetheless, the chemical search space is (Fig. 1b). We use informatics-based methods to screen the chemical space and downselect 242 compositions that show greater promise for NCS

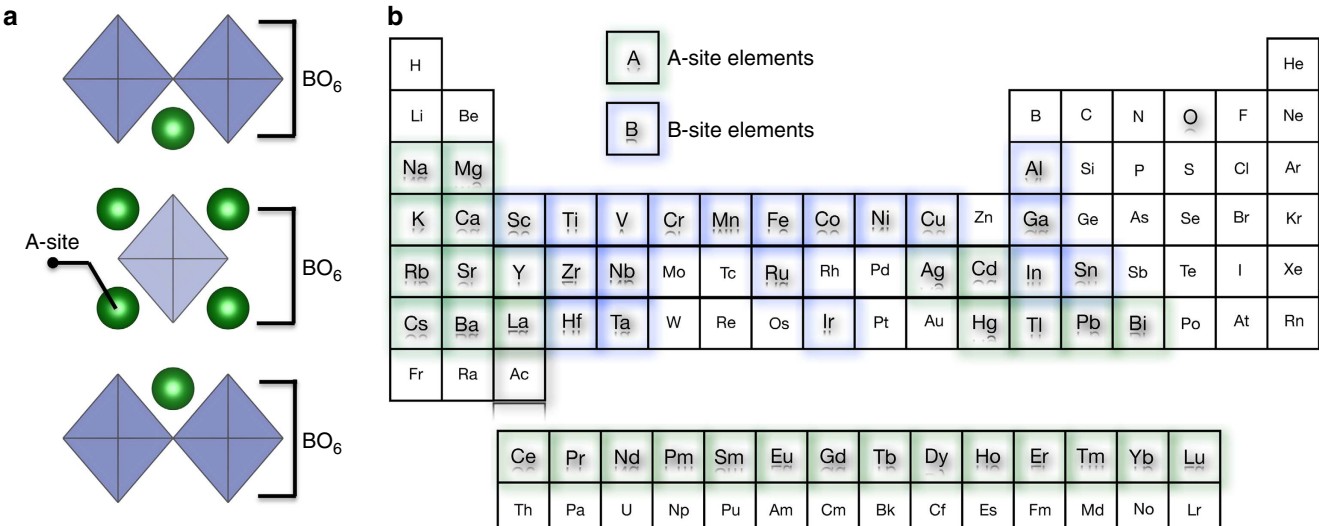

**Figure 1 | Octahedral connectivity of $n=1$ RP oxides and the chemical search space. (a)** The $n=1$ RP phase has a single layer of octahedra that are connected in two dimensions, shown within brackets, whereas there is no connectivity in the third dimension. **(b)** Periodic table showing the potential 30 A-site and 19 B-site elements that occupy the $n=1$ RP phase. In principle, there are more than 19 B-site elements when we also consider the multiple valence states of certain elements (for example, Mn, Fe, Co, Ni and so on). This defines the chemical space for our informatics approach.

ground states. The potential for discovering novel NCS $n=1$ RP compounds has key implications in technological applications that require a broad range of functionalities including high-temperature piezoelectricity, tunable bandgaps, improper ferroelectricity, multiferroicity and thermoelectricity. We focus in detail on the design of NaRSnO$_4$ stannates and NaRRuO$_4$ ruthenates (where $R =$ La, Pr, Nd, Gd or Y) that were predicted to have NCS ground state structures from informatics and subsequently validated by density-functional theory (DFT). For the stannates, which are candidate materials for sensors and transparent conducting oxides[18], we find two energetically competing NCS ground state phases: $P\bar{4}2_1m$ (piezo-active) and $P2_12_12$ (chiral and piezo-active). We calculate their electronic bandgaps in the $P\bar{4}2_1m$ crystal symmetry using hybrid exchange-correlation functionals, finding optical transparency in the visible light regime. We also compute their piezoelectric responses that show a dependence on $R$-cation size. In sharp contrast, the NCS NaRRuO$_4$ are magnetic with metallic, half-metallic or insulating electronic structures. Their ground state is determined to be either piezo-active with $P\bar{4}2_1m$ symmetry when $R =$ La, Pr and Nd or polar with $Pca2_1$ symmetry when $R =$ Gd and Y. Moreover, there is a transition from ferromagnetic metallic ($R =$ La) or half-metallic ($R =$ Pr, Nd) to antiferromagnetic insulating ($R =$ Gd, Y) character as a function of $R$-cation size. Therefore, these bulk ruthenates are predicted to belong to the intriguing class of NCS metals[19,20] and half-metals with piezo-active symmetries or antiferromagnetic insulators with polar symmetry (that is, multiferroics). Last, we also test our predictions for an additional nine new compounds with different cations occupying the B-sublattice of the RP structure (shown in Fig. 1a). Among them, seven were validated to have an NCS ground state structure—NaLaZrO$_4$, NaLaHfO$_4$, KBaNbO$_4$, NaLaIrO$_4$, NaCaTaO$_4$, SrYGaO$_4$ and SrLaInO$_4$. These results establish our computational framework as a powerful tool for crystal symmetry classification, structure-based property design and control.

## Results

**Approach**. Our search for NCS oxides relies on a multifaceted theoretical approach, which reformulates the discovery objective into identifying structure—chemistry interrelationships (as shown in Fig. 2). The design strategy focuses on three key criteria obtained by subdividing the design process into unique objectives with specific tasks:

- Structural: How can the atomic structure, or configuration of oxygen octahedra BBUs, be designed to support the desired interaction?
- Chemical: Which combinations of chemistries will promote that structural configuration?
- Stability: Is the proposed composition the global ground state?

Following classification learning from informatics and evaluation of the energetic stability from first principles methods, the final design relies on response optimization by leveraging additional degrees of freedom to further promote the targeted behaviour. Some of the strategies include searching for microscopic mechanisms and external conditions (such as epitaxial strain) to energetically stabilize those geometries. We note that this paper is a significant advancement from the earlier work of Balachandran et al.[15] where the emphasis was on enumerating symmetry guidelines.

**Group theory**. In an earlier work, Balachandran et al.[15] formulated symmetry guidelines for exploring and designing

NCS phases in the $n=1$ RP structures based on group theory. Therefore, we discuss only the key results here. Starting from the centrosymmetric (CS) aristotype structure (shown in Fig. 3a), various symmetry-allowed cooperative atomic displacements (also referred to as 'shuffles') were enumerated that transform the aristotype CS structure to a NCS structure of lower symmetry. Particularly, the focus was on CS→NCS phase transitions that are second order or weakly first order, where the symmetry-lowering distortions arise from (i) non-polar octahedral distortions (tilting or rotations) due to phonon softening at the zone boundaries in the BZ of the $I4/mmm$ space group, (ii) A/A′ cation ordering, (iii) the interplay between two or more octahedral distortions and (iv) the interplay between octahedral distortions and A/A′ cation ordering. The necessity to search for alternative routes to breaking inversion symmetry was motivated by the fact that NCS phases are seldom seen in $n=1$ RPs, which has been explained by the disconnected octahedral layers destroying the coherency required for cooperative off-centring displacements, and thus ferroelectricity[21].

Balachandran et al.[15] found three important symmetry guidelines (given in the rows of Table 1) for lifting parity in the $n=1$ RP structures. Note that all involve A/A′ cation ordering (Fig. 3b) that transform as irreducible representation (irrep) $M_3^-$ and couple with octahedral rotations or tilting (as shown in Fig. 3c–e). The structural attributes may be satisfied by any of the following approaches:

Route 1: Out-of-phase octahedral tilting that transform as irrep $X_3^+$ with order parameter direction (OPD) $(\eta_1,\eta_1)$, which on superposition with irrep $M_3^-$ $(\eta_1)$ would yield a piezoelectric ($P\bar{4}2_1m$) space group (Fig. 3c).

Route 2: Out-of-phase tilting that transform as irrep $X_3^+$ with OPD $(\eta_1,\eta_2)$ on superposition with $M_3^-$ $(\eta_1)$ would yield a chiral ($P2_12_12$) and piezo-active space group (Fig. 3d).

Route 3: Coupled irrep $X_2^+ \oplus X_3^+$ with OPD $(0,\eta_1;\eta_2,0)$ when superposed with irrep $M_3^-$ $(\eta_1)$ would yield a polar ($Pca2_1$) space group (Fig. 3e), where the matrix elements of $X_2^+$ and $X_3^+$ irreps accommodate atomic displacements that correspond to Jahn-Teller-like distortions and out-of-phase tilting, respectively.

Note that there is another type of A/A′ cation ordering, transforming as irrep $\Gamma_3^-$, which lifts inversion solely from the ordering (we refer to it as the trivial case). However, we do not consider $\Gamma_3^-$ A/A′ cation ordering here. Therefore, the key materials design question is: What combinations of chemical elements from the vast chemical space would stabilize these NCS phases? We address this question using materials informatics.

**Materials informatics**. In Fig. 4, we show the frequency of occurrence of experimentally known crystal symmetries in the bulk $n=1$ RPs. We report only the low temperature crystal symmetries in Fig. 4 and do not explicitly consider temperature dependence of the crystal structures in our informatics analysis. Our definition of low temperature includes experimentally observed structures $\leq 300$ K. Some RP compounds also undergo structural transformation at a much lower temperature (for example, La$_2$NiO$_4$ (ref. 22)). Under such circumstances, we take the lower temperature crystal structure to be our label for informatics. This simplification was necessary because 0 K DFT calculations are used to validate the informatics-based predictions. Balachandran et al.[15] showed that as the temperature increases, the propensity for forming high-symmetry phases also increases. We anticipate those results to hold here.

Our literature survey shows that $\sim 45\%$ of the compositions are undistorted (denoted as $\phi$ in Fig. 4). Similarly, there are also a significant number of compositions that undergo symmetry-lowering distortions, albeit preserving the spatial inversion

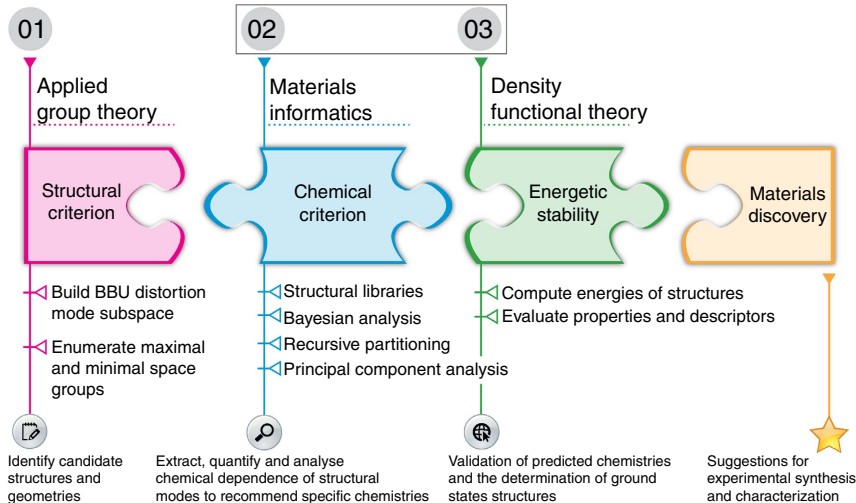

**Figure 2 | Predictive materials discovery framework.** Synergistic integration of applied group theory, materials informatics and *ab initio* electronic structure calculations for designing novel functional materials. Applied Group Theory determines the geometric rules, uncovers the crystallographic symmetry restrictions and then subsequently shows how to lift them to achieve NCS structures for a given crystal structure topology. Materials informatics uses the data from experiments, features (such as orbital radii) that capture the chemical trends in the constructed data set and statistical inference tools to extract QCSR that guides selection of chemical compositions. DFT calculations validate the predictions from materials informatics. We then recommend the validated chemical compositions for experimental synthesis and characterization, eventually leading to its discovery. Experimentally synthesized compositions augment the training set for a second materials informatics iteration and the process repeats until desired materials are discovered[14]. In this paper, we focus on computational tasks 2 and 3 (boxed).

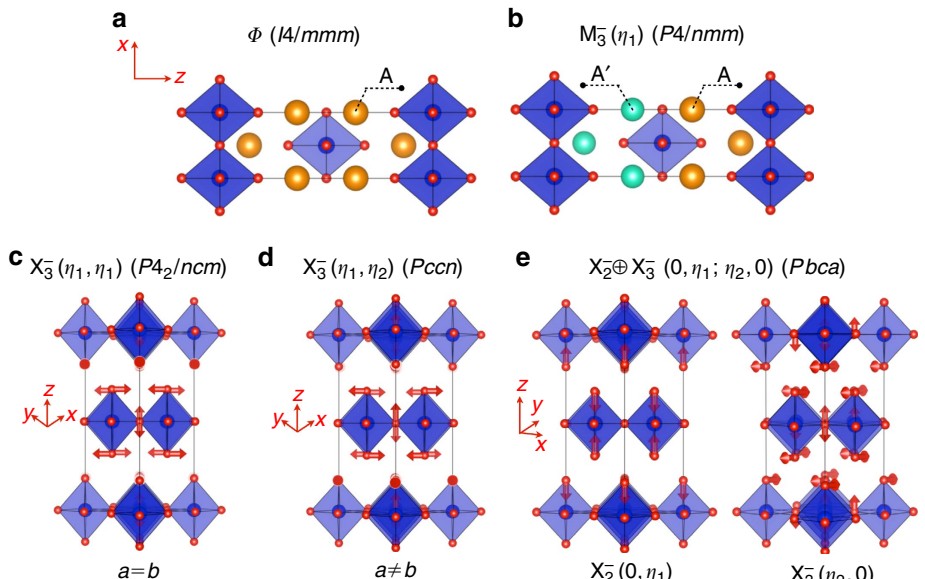

**Figure 3 | A/A′ cation ordering and octahedral tilting in the $n = 1$ RPs for NCS materials design.** (**a**) High-symmetry aristotype structure ($\phi$, $I4/mmm$). (**b**) One of the A/A′ cation ordering schemes (irrep: $M_3^-$ ($\eta_1$); space group (s.g.): $P4/nmm$). (**c**) Out-of-phase octahedral tilting (oxygen displacements indicated using arrows) (irrep: $X_3^-$ ($\eta_1,\eta_1$); s.g.: $P4_2/ncm$) and lattice constants $a$ and $b$ are of equal length. (**d**) Out-of-phase octahedral tilting (irrep: $X_3^-$ ($\eta_1,\eta_2$); s.g.: $Pccn$) and lattice constant $a \neq b$. (**e**) Coupled distortions (irrep: $X_2^+ \oplus X_3^-$ ($0,\eta_1;\eta_2,0$); s.g.: $Pbca$), where $X_2^+$ ($0,\eta_1$) and $X_3^-$ ($\eta_2,0$) represent Jahn–Teller-like out-of-plane compression and out-of-phase octahedral tilting, respectively.

symmetry. One of the key observations from Fig. 4 is that there are only nine compounds with NCS space groups that conform with our chemical search space (Fig. 1b). In the literature, the family of cation-ordered NaRTiO$_4$ and LiRTiO$_4$ (found only recently), where $R$ = La, Nd, Dy, Gd, Sm, Ho, Eu and Y, have been experimentally shown[16,17] to have the piezoelectric $P\bar{4}2_1m$ space group [$X_3^+ \oplus M_3^-$ ($\eta_1,\eta_1;\eta_1$)]. The nominal electronic configuration of Ti$^{4+}$ in these compounds is $d^0$. The coupling between TiO$_6$ octahedral tilting (that transform as irrep $X_3^+$ ($\eta_1,\eta_1$) as shown in Fig. 3c) and Li/$R$ or Na/$R$ cation ordering

(that transform as irrep $M_3^-$ ($\eta_1$) as shown in Fig. 3b) lifts the inversion symmetry—in accordance with Route 1. The only other experimentally known polar $n = 1$ RP oxide is the A- and B-site-ordered (LaSr)(Li$_{0.5}$Ru$_{0.5}$)O$_4$ compound, which is reported in the NCS $Imm2$ space group[23]. In this compound, a combination of A-site and B-site cation ordering work in concert to lift the inversion symmetry. In addition to these compounds, Pb$_2$TiO$_4$, Ca$_2$IrO$_4$, Sn$_2$SnO$_4$, cation-ordered La$A$NiO$_4$ ($A$ = Sr, Ca and Ba) LaSrAlO$_4$ and LaSrMnO$_4$ have also been theoretically predicted to have NCS structures[15,24–28]; however, these results have not

**Table 1 | Irreps, OPDs, SGs and mode representation of distorted structures arising from rotational modes ($X_2^+$ and $X_3^+$) and A-site cation ordering ($M_3^-$).**

| Irreps | OPD | SG | MR |
|---|---|---|---|
| $X_3^+ \oplus M_3^-$ | $(\eta_1,\eta_1;\eta_1)$ | $P\bar{4}2_1m$ | Rotation + ACO |
| $X_3^+ \oplus M_3^-$ | $(\eta_1,\eta_2;\eta_1)$ | $P2_12_12$ | Rotation + ACO |
| $X_2^+ \oplus X_3^+ \oplus M_3^-$ | $(0,\eta_1;\eta_2,0;\eta_1)$ | $Pca2_1$ | Rotations + ACO |

ACO, A-site cation ordering; MR, mode representation; OPD, order parameter direction; SG, space group; $\oplus$, coupled distortions.

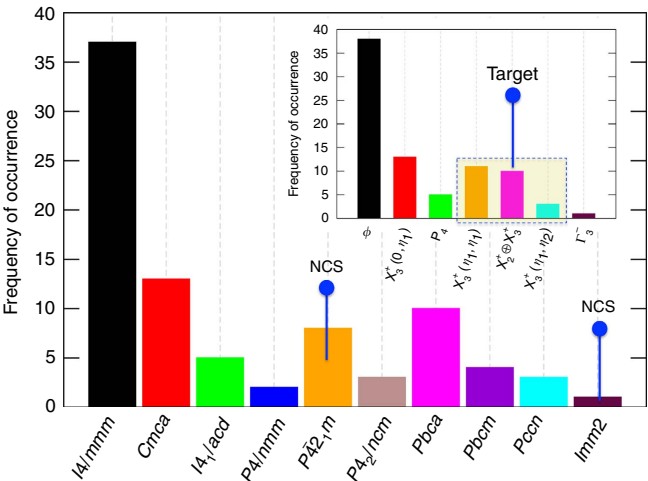

**Figure 4 | Distribution of experimentally known RP oxides.** Our survey resulted in a total of 84 compounds, which we note represents only a small fraction of the overall combinations of hypothetically feasible chemistries. Except for the nine compounds indicated in space groups $P\bar{4}2_1m$ and $Imm2$, there are no other experimental reports of NCS phases in $n=1$ RP oxides. Inset: The space groups are transformed into their corresponding irreducible representations (irreps) and A/A' cation ordering is not explicitly considered. The symbol $\phi$ denotes no octahedral rotation or tilting. Irreps that we target for NCS materials design are indicated using the dotted rectangle in the inset.

been experimentally verified. Recently, the metastable $Ca_2IrO_4$ was epitaxially grown on a $YAlO_3$ substrate in the $n=1$ RP phase using pulsed laser deposition[29]. However, the authors did not report its crystal symmetry. Therefore, we do not consider these chemistries in our informatics analysis.

In the family of $n=1$ RPs with relatively simple stoichiometries such as $AA'BO_4$, where A and A' are two chemical species (similar or dissimilar) occupying the A-site and B is a cation with 6-fold octahedral coordination, there are $\sim 3,200$ potential chemical compositions that satisfy crystal chemistry and stoichiometric guidelines (for example, charge neutrality), and therefore are, in principle, amenable for experimental synthesis. However, only 3% have been experimentally synthesized, and among these, only nine have NCS phases. The objective of our informatics analysis is to utilize statistical inference and machine learning (ML) methods for establishing quantitative chemistry-symmetry relationships (QCSR) of known materials in Fig. 4. These QCSRs, in turn, serve as a guide to rapidly screen the vast chemical space and identify new, previously unexplored compositions that favour the distortions given in the Table 1.

*Data set.* In our ML approach, we build a data set of experimentally known materials that includes both CS and NCS structures. Even though our computational design focuses on $AA'BO_4$ stoichiometries, our training data set includes RP compositions that deviate from the $AA'BO_4$ stoichiometry (see data set in the Supplementary Information). We describe each $n=1$ RP composition uniquely in terms of its crystal symmetry or irrep (referred to as 'class label' in the ML jargon) and a set of features. We use Waber–Cromer orbital radii as features for ML[30]. Orbital radii and distortion modes have been utilized in the past for predicting structures and formabilities of complex oxides[31,32]. Our ML objective is to build a classification model that predicts crystal symmetries or irrep labels from orbital radii. All 83 experimentally known RP chemical compositions (after removing $(LaSr)(Li_{0.5}Ru_{0.5})O_4$, because we do not consider the element Li in our chemical space, see Fig. 1b) were written in the simplified $A_2BO_4$ stoichiometric form, where the A- and B-sites can have two or more elements with partial site occupancies. We used a total of 12 and 10 orbital radii features to describe the A- and B-sites, respectively. If there were two or more elements occupying either the A- or B-sites, then linear combinations weighted by their relative stoichiometric proportions were used to build the features.

We constructed two data sets for classification learning that uses: (i) space groups as class labels (an obvious choice) and (ii) irreps corresponding to octahedral tilting, rotations, or lack thereof as class labels. Here, we focus mainly on the ML results from the latter data set (case (ii)) that uses irreps as class labels, which allows us to elegantly isolate octahedral rotations or tilting from cation ordering. As a result, we can group or combine two space groups under the same label. For example, we combine compositions with the $I4/mmm$ and $P4/nmm$ space group together (under the label, $\phi$), because in both cases there are no

octahedral rotations or tilting. One of the key differences between $I4/mmm$ and $P4/nmm$ is that in $P4/nmm$ the A-site Wyckoff orbit is split into two unique crystallographic sites[15]. Similarly, we can combine space groups $P\bar{4}2_1m$ and $P4_2/ncm$ into a single irrep, $X_3^+$ $(\eta_1,\eta_1)$. Such data transformation reduces the number of unique class labels from 9 to 7 (see inset in Fig. 4) for classification learning. The main disadvantage with such grouping is that our QCSR model now cannot distinguish between ordered and disordered structures. This should not affect our NCS materials design goal because of advancements in the nonequilibrium synthesis and processing of these oxides. Recently, there have been experimental demonstrations of layer-by-layer growth of A/A' cation-ordered $n=1$ RPs using molecular beam epitaxy with unprecedented control[33]. We also tested the predictive power of our ML models by intentionally leaving out 14 compounds during training (which reduces the size of our training set from 83 to 69 compounds). One of our informatics goals is to validate whether our classification learning can identify the labels correctly for the left out compounds, before using them for making new predictions.

Even after reducing the number of unique class labels from 9 to 6 (since there is only one chemical composition with irrep $\Gamma_3^-$, which we do not consider for ML), we must still address the problem of class imbalance, where some irrep class labels are found more frequently than others. This kind of class imbalance is problematic for ML. To test the implications of class imbalance, we trained a decision tree classification model using the imbalanced data set and found that compositions with space group $Pccn$ or $X_3^+$ $(\eta_1,\eta_2)$ were 100% misclassified. As shown in Table 1 and Fig. 3, $Pccn$ or $X_3^+$ $(\eta_1,\eta_2)$ is one of the desired class labels for designing NCS materials. Therefore, the class-imbalance problem must be addressed.

A number of methods have been developed in the computer science and artificial intelligence literature to overcome the class-imbalance problem[34,35]. Some of them include: oversampling (that is, randomly duplicating instances of the under-represented

class category), undersampling (random removal of instances of the most frequently occurring class) and interpolation schemes. In this work, we utilize an oversampling scheme referred to as synthetic minority class oversampling technique (or SMOTE)[34], in which the under-represented class labels are oversampled by creating 'synthetic' examples of extra or fictitious training data points from the original imbalanced data. It is based on a $k$-nearest-neighbour analysis and one of its main advantages (relative to other algorithms) is that the extra data points, in principle, informs the ML models to create larger and less specific decision regions. Additional details about the algorithm are described in the Methods section.

We took the data set that contained irreps as class labels and applied SMOTE to construct synthetic data points for the two irrep labels, $P_4$ and $X_3^+$ ($\eta_1,\eta_2$). We created a total of three and six synthetic data points for the under-represented $P_4$ and $X_3^+$ ($\eta_1,\eta_2$) labels, respectively. Our training data set size now increased to 78 compounds (69 originally + 9 from SMOTE) for classification learning. We confirmed using principal component analysis (PCA) that SMOTE did not affect our data manifold (Supplementary Fig. 1).

*Data preprocessing.* Our NCS materials design is initiated by exhaustively enumerating, at first, all possible $AA'BO_4$ combinations that satisfy crystal chemistry and stoichiometric rules (for example, charge neutrality). As noted before, we use Waber–Cromer orbital radii as features. We then augment this exhaustive data set with the 78 $n = 1$ RPs. Note that at this point, we do not include the irrep class labels in our data set. Now, we have a total of 3,253 chemical compositions and 22 orbital radii features.

We autoscaled the data (normalized to zero mean and unit variance) and applied PCA, which constructs linear combinations of weighted contributions of orbital radii (see Supplementary Figs 2 and 3). In a recent work, Balachandran *et al.*[36] showed that in a data set containing orbital radii as features, PCA removes redundancy of information, reduces data dimensionality and constructs physically meaningful linear combinations of orbital radii (see Supplementary Note 1). In addition, principal components (PCs) are also independent of one another (assuming Gaussian or Normal distribution). After PCA, we reduced the dimensionality of our data set from 22 orbital radii features to 8 PCs, which together capture > 90% of total variance in the data set. We then identify and isolate 78 chemical compositions for which the irrep labels are experimentally

known; we refer to this data set as the training set. The remaining compositions are referred to as the 'virtual set' defining the vast chemical search space yet to be explored for new NCS materials design.

*Classification learning.* We utilized the J48 decision tree classification learning algorithm, as implemented in WEKA, for establishing QCSR[37,38]. The reasons for choosing the J48 algorithm are discussed in the Methods section. We constructed five bootstrapped samples of 78 compositions each from the original training set. We then trained the decision tree algorithm using the five bootstrapped samples and constructed five decision tree models (Supplementary Figs 4–8). The classification accuracies for the five decision tree models were evaluated on the training data set and by 10-fold cross-validation. The results are given in Supplementary Table 1 and Supplementary Note 2. The average classification accuracy from the five bootstrapped decision trees using the 10-fold cross-validation is ∼80%. These results indicate that more accurate QCSR models could potentially be formulated either through alternative feature selection methods[39] or by utilizing other (kernel-based) ML algorithms (which we do not address here). Furthermore, we also tested our decision trees to determine whether they could correctly identify the irrep labels for 14 compounds, which were intentionally held out during the training process. Results are given in Table 2. Our ensemble of decision trees correctly labelled with ≥60% accuracy (except for $YSrCrO_4$ and $Ca_2CrO_4$) 12 out of 14 compounds in the independent test set, giving confidence in our classification learning.

Using the five bootstrapped decision trees, we screened a total of 3,175 compositions in the virtual set and filtered 242 new compositions that showed potential for NCS ground state structures. At this stage, we retained only those compositions that were identified to be NCS, that is, belonging to either $X_3^+$ ($\eta_1,\eta_1$), $X_3^+$ ($\eta_1,\eta_2$) or $X_2^+ \oplus X_3^+$ ($0,\eta_1;\eta_2,0$), by at least three out of the five decision trees. We then created additional filters to remove data points that contained (i) toxic elements, such as Pb, Hg and Cd, (ii) compositions where both A and A' sites were occupied by the same element and (iii) compositions with A or A' site elements that were not part of the original training data set (for example, Cs, Rb, Tl, Ag and Mg).

We note that some disagreement is expected between our predictions and experiments (or calculations), particularly when concerned with the transition metal elements whose valence state falls within the strong electron correlations regime (for example,

---

**Table 2 | A comparison between experimental and predicted irreps to independently validate the classification models.**

| RP oxides | Experimental irrep | Predicted irrep | Prediction accuracy (in %) |
|---|---|---|---|
| $CaSrRuO_4$ (ref. 74) | $P_4$ | $P_4$ | 60 |
| $LaSrFeO_4$ (ref. 75) | $\phi$ | $\phi$ | 100 |
| $LaSrCoO_4$ (ref. 76) | $\phi$ | $\phi$ | 100 |
| $NdSrCoO_4$ (ref. 76) | $\phi$ | $\phi$ | 100 |
| $GdSrCoO_4$ (ref. 76) | $\phi$ | $\phi$ | 100 |
| $LaSrCrO_4$ (ref. 77) | $\phi$ | $\phi$ | 100 |
| $YCaCrO_4$ (ref. 77 | $X_3^+$ ($\eta_0,\eta_1$) | $X_3^+$ ($\eta_0,\eta_1$) | 80 |
| $YSrCrO_4$ (ref. 77) | $X_3^+$ ($\eta_1,\eta_2$) | $\phi$ | 0 |
| $SmCaCrO_4$ (ref. 78) | $X_3^+$ ($\eta_0,\eta_1$) | $X_3^+$ ($\eta_0,\eta_1$) | 100 |
| $LaCaFeO_4$ (ref. 79) | $X_3^+$ ($\eta_0,\eta_1$) | $X_3^+$ ($\eta_0,\eta_1$) | 80 |
| $Ca_2CrO_4$ (ref. 80) | $P_4$ | $P_4$ and $X_3^+$ ($\eta_0,\eta_1$) | 40 |
| $NaDyTiO_4$ (ref. 16) | $X_3^+$ ($\eta_1,\eta_1$) | $X_3^+$ ($\eta_1,\eta_1$) | 100 |
| $NaSmTiO_4$ (ref. 16) | $X_3^+$ ($\eta_1,\eta_1$) | $X_3^+$ ($\eta_1,\eta_1$) | 100 |
| $NaHoTiO_4$ (ref. 16) | $X_3^+$ ($\eta_1,\eta_1$) | $X_3^+$ ($\eta_1,\eta_1$) | 100 |

Prediction accuracy (in %) is the ratio of the number of trees that correctly predicted the irrep label to the total number of trees ( = 5) used for prediction. All experimentally reported compounds have disordered A-site arrangement. In $Ca_2CrO_4$, our classifier predicts with 40% confidence that both $P_4$ and $X_3^+$ ($\eta_0,\eta_1$) labels are equally likely and experimentally, $P_4$ is observed.

**Table 3 | Full list of 242 predicted $AA'BO_4$ RP compounds from classification learning that show propensity towards NCS structures.**

| B-cation | [A; A′ cation combinations] |
|---|---|
| $Ga^{3+}$ | [A = Sr; A′ = Y, Er, Tm and Yb]<br>[A = Ba; A′ = Bi, La, Ce, Pr, Nd, Pm, Sm, Eu, Gd, Tb, Dy, Ho, Er, Tm, Yb and Lu] |
| $In^{3+}$ | [A = Ca; A′ = Bi, La, Ce, Pr, Nd, Pm, Sm, Eu, Gd, Tb, Dy, Ho, Er, Tm, Yb and Lu]<br>[A = Sr; A′ = Y, Bi, La, Ce, Pr, Nd, Pm, Sm, Eu, Gd, Tb, Dy, Ho, Er, Tm, Yb and Lu]<br>[A = Ba; A′ = Y and Bi] |
| $Ti^{4+}$<br>$Zr^{4+}$ | [A = Na; A′ = Bi, Ce, Pm, Tm, Yb and Lu]<br>[A = Na; A′ = Y, Bi, La, Ce, Pr, Nd, Pm, Sm, Eu, Gd, Tb, Dy, Ho, Er, Tm, Yb and Lu]<br>[A = K; A′ = Bi, La, Ce, Pr, Nd, Pm, Sm, Eu, Gd, Tb, Dy, Ho, Er, Tm, Yb and Lu]<br>[A = Ca, Sr; A′ = Ba] |
| $Ru^{4+}$ | [A = Na; A′ = Y, Bi, La, Ce, Pr, Nd, Pm, Sm, Eu, Gd, Tb, Dy, Ho, Er, Tm, Yb and Lu]<br>[A = K; A′ = Bi, La, Ce, Pr, Nd, Pm, Sm, Eu, Gd, Tb, Dy, Ho, Er, Tm, Yb and Lu]<br>[A = Ca, Sr; A′ = Ba] |
| $Sn^{4+}$ | [A = Na; A′ = Y, Bi, La, Ce, Pr, Nd, Pm, Sm, Eu, Gd, Tb, Dy, Ho, Er, Tm, Yb and Lu]<br>[A = K; A′ = Bi, La, Ce, Pr, Nd, Pm, Sm, Eu, Gd, Tb, Dy, Ho, Er, Tm, Yb and Lu]<br>[A = Ca; A′ = Ba] |
| $Hf^{4+}$ | [A = Na; A′ = Y, Bi, La, Ce, Pr, Nd, Pm, Sm, Eu, Gd, Tb, Dy, Ho, Er, Tm, Yb and Lu]<br>[A = K; A′ = Bi, La, Ce, Pr, Nd, Pm, Sm, Eu, Gd, Tb, Dy, Ho, Er, Tm, Yb and Lu]<br>[A = Ca; A′ = Ba] |
| $Ir^{4+}$ | [A = Na; A′ = Y, Bi, La, Ce, Pr, Nd, Pm, Sm, Eu, Gd, Tb, Dy, Ho, Er, Tm, Yb and Lu]<br>[A = K; A′ = Bi, La, Ce, Pr, Nd, Pm, Sm, Eu, Gd, Tb, Dy, Ho, Er, Tm, Yb and Lu] |
| $Nb^{5+}$ | [A = Na; A′ = Ca, Sr and Ba]<br>[A = K; A′ = Ca and Ba] |
| $Ta^{5+}$ | [A = Na; A′ = Ca, Sr and Ba]<br>[A = K; A′ = Ca and Ba] |

NCS, noncentrosymmetric; RP, Ruddlesden-Popper.

$Ti^{3+}$, $Cr^{3+}$, $V^{3+}$, $Mn^{3+}$ and so on), mainly because there were very few instances of chemical compositions with these transition metal cations in our training set. Our refined results, after screening through various filters and removing chemical compositions that could fall in the strongly correlated regime, included a total of 242 new chemical compositions that show promise for NCS structures.

The following octahedral B-site cations in the virtual set are predicted to have NCS structures in the $n = 1$ RP oxides: $Ga^{3+}$, $In^{3+}$, $Ti^{4+}$, $Zr^{4+}$, $Ru^{4+}$, $Sn^{4+}$, $Hf^{4+}$, $Ir^{4+}$, $Nb^{5+}$ and $Ta^{5+}$. We could also exclude $In^{3+}$, because of the experimental difficulties in forming $n = 1$ RP structures using equilibrium synthesis and processing techniques[40] (although we do not preclude stabilizing In-based $n = 1$ RPs using non-equilibrium methods). The chemical compositions for all predicted NCS materials are listed in Table 3. Additional details can be found in

Supplementary Table 2, Supplementary Note 3 and the data sets can be downloaded from ref. 41. To summarize, using informatics we identified 242 new $n = 1$ RP chemical compositions with potential for NCS crystal structures, which significantly expands the chemical space of NCS $n = 1$ RP oxides ($\sim$25-fold increase).

**Density-functional theory.** On the basis of the group theory and materials informatics analysis, we first validate our predictions by assessing the energetic stability component (Task 3 in Fig. 2) for ten downselected $NaRSnO_4$ and $NaRRuO_4$ compounds, where $R$ is a rare-earth element ($R$ = La, Pr, Nd, Gd and Y) using DFT calculations. In our calculations, $Na^{1+}$ and $R^{3+}$ cations were ordered in accordance with the irrep label $M_3^-$ ($\eta_1$), as shown in Fig. 3b. To the best of our knowledge, no previous experimental or theoretical data exists for either $NaRSnO_4$ or $NaRRuO_4$ compounds. In addition, stannates have implications in the design of transparent conducting oxides[18] and ruthenates are potential materials for investigating metal–insulator transitions[42].

We choose especially $NaRSnO_4$ and $NaRRuO_4$ for validation, motivated (albeit naively) by the adaptive design paradigm[14], where the objective is to iteratively improve the predictions of the classification model. Typically, the improvements are made by choosing chemical compositions for experiment that show promising characteristics (such as NCS crystal classes as discussed here), yet have large uncertainties. Here, $NaRSnO_4$ and $NaRRuO_4$ satisfy these criteria, because the predictions from the five decision trees were $X_2^+ \oplus X_3^+$ (NCS), $X_3^+$ ($\eta_1, \eta_2$) (NCS), $X_3^+$ ($0, \eta_1$) (CS), $X_2^+ \oplus X_3^+$ (NCS) and $X_3^+$ ($\eta_1, \eta_2$) (NCS), corresponding to $Pca2_1$ (polar), $P2_12_12$ (chiral), $Pbcm$ (centrosymmetric), $Pca2_1$ (polar) and $P2_12_12$ (chiral) space groups, respectively. Four out of the five decision trees predict these compounds to have a chiral or polar structure, making them promising NCS candidates, yet the irrep labels or space groups are different, indicating uncertainty. Furthermore, with stannates the nominal electronic configuration of $Sn^{4+}$ ($4d^{10}$) is different from that of SOJT-cation $Ti^{4+}$ ($3d^0$), thereby presenting an interesting case for comparison between the two B-site octahedral cations. The Shannon ionic radii for $Sn^{4+}$ and $Ti^{4+}$ in the six-fold coordination are 0.69 and 0.605 Å, respectively[43], making their ionic sizes within the hard-sphere model also different. Similarly, ruthenates (with Ru in nominally $4+$ ionic state) have partially filled $4d$ electrons with four electrons occupying the $t_{2g}$ orbital manifold and are quite distinct from the $3d^0$ titanates.

*Stannates.* We performed full structural relaxations for $NaRSnO_4$ (where $R$ = La, Pr, Nd, Gd and Y) within the generalized gradient approximation (cf. Methods). The phonon dispersions are given in Supplementary Fig. 9, from which we identify a common set of six candidate crystal symmetries from 'freezing in' the imaginary phonon modes of the high-symmetry paraelectric reference phase ($P4/nmm$) for determining the ground state structure. They include $Pmn2_1$, $Pc$, $P\bar{4}2_1m$, $P\bar{4}2m$, $I\bar{4}2m$ and $Pnma$. In addition to these six crystal symmetries, we also considered three more symmetries, namely $P2_12_12$, $Pbcm$ and $Pca2_1$, as recommended by ML to unambiguously confirm the ground state. Therefore, in total, we considered nine distorted candidate structures. The total energy data from DFT calculations is given in Table 4, which shows that all stannates exhibit a strong energetic competition between the NCS piezoelectrically active $P\bar{4}2_1m$ [$X_3^+$ ($\eta_1, \eta_1$)] and chiral $P2_12_12$ symmetries [$X_3^+$ ($\eta_1, \eta_2$)]. We find that the total energy difference is $< 0.1$ meV per f.u. (Table 4) between the two NCS phases. A closer examination of the two converged crystal structures revealed that they differ mainly in the in-plane lattice parameters (in $P\bar{4}2_1m$ $a = b$, whereas in $P2_12_12$ $a \neq b$ and this is shown in Fig. 3c,d, respectively). Furthermore, in $P2_12_12$ the in-plane lattice constant

$a$ was found to be not equal to $b$ only in the fourth or fifth decimal point. Therefore, we assign the ground state structure to be NCS $P\bar{4}2_1m$ space group for the stannates. We conclude from our DFT calculations that the RP stannates are NCS, in good agreement with the insights from ML and the inversion symmetry is broken due to the coupled action of $SnO_6$ oxygen octahedral tilting and Na/$R$ cation ordering (Route 1).

We then computed the bandgaps ($E_g$) for each of the compounds using the HSEsol exchange-correlation functional (which often more accurately reproduces experimental results[44]) and found them to be in the range 4.3 to 4.5 eV (Table 5), similar to $Ba_2SnO_4$ ($E_g = 4.41$ eV)[18]. The amount of exact exchange used in the calculations was tuned using the known experimental bandgap of $BaSnO_3$ (ref. 45).

We next computed the piezoelectric strain coefficients ($d_{ij}$) for each compound in $P\bar{4}2_1m$ space group (Fig. 5); the $d_{ij}$ response is marginally smaller than that reported for the titanates[16], but follows the same trend (increasing with decreasing atomic radius, up to $R = $ Gd and then decreases).

*Ruthenates.* All DFT calculations were performed using the spin-polarized DFT + $U$ method, where an effective Hubbard-$U$ of 1.5 eV was used to treat the correlated Ru 4$d$ electrons (cf. Methods). The phonon dispersions are given in Supplementary Fig. 10 and show some similarities with the stannates. We explored a total of nine distorted crystal symmetries to determine the ground state (six from phonon calculations and three from ML). The total energies from DFT + $U$ for NaRRuO$_4$ in different crystal symmetries and ferromagnetic spin order are given in Table 4; the ground state is determined to be NCS for NaLaRuO$_4$, NaPrRuO$_4$ and NaNdRuO$_4$ with two competing structures, $P2_12_12$ and $P\bar{4}2_1m$. Moreover, in the $P2_12_12$ symmetry, $a$ was found to be not equal to $b$ only at the fourth decimal point (similar to the stannates). We also performed additional DFT + $U$ calculations for the top two lowest energy structures (namely $P\bar{4}2_1m$ and $Pca2_1$), where we now impose antiferromagnetic spin order on the in-plane Ru atoms (shown schematically in Supplementary Fig. 11). The total energy results are given in Table 6, from which we conclude that the NCS

$P\bar{4}2_1m$ space group with ferromagnetic $Ru^{4+}$–$O^{2-}$–$Ru^{4+}$ interactions is the likely ground state for these compounds (Route 1).

In the case of NaGdRuO$_4$ and NaYRuO$_4$, the ground state structure is also determined to be NCS, but in polar $Pca2_1$ crystal symmetry (see Table 4). Furthermore, in both NaGdRuO$_4$ and NaYRuO$_4$, the $Pca2_1$ structure with in-plane antiferromagnetic $Ru^{4+}$–$O^{2-}$–$Ru^{4+}$ interactions (Supplementary Fig. 11) were found to be 1.44 and 5.54 meV per atom lower in energy, respectively, than that for the ferromagnetic structures. The total energy data along with Ru-atom magnetic moments are given in Table 6. Thus, we predict NaGdRuO$_4$ and NaYRuO$_4$ to have polar $Pca2_1$ ground state structures (Route 3) with antiferromagnetic spin order.

We also calculated the electronic band structures for all five NaRRuO$_4$ in their respective ground states. The results are shown in Supplementary Fig. 11. We find that NaLaRuO$_4$ is metallic with bands crossing the Fermi level in both the spin-up and spin-down electron channels. On the other hand, the NaPrRuO$_4$ and NaNdRuO$_4$ are found to be half-metals, that is, bands cross the Fermi level only in the spin-down channel and a gap appears for the spin-up channel. Moreover, the size of the gap increases as the rare-earth cation size decreases. This occurs because the relative amplitude of $RuO_6$ octahedral tilting also increases with decreasing rare-earth cation size, impacting the electronic bandwidths of the Ru-$t_{2g}$ orbitals. Note that this is not the first time ferromagnetic metals or half-metals are reported in ruthenium-based oxides[46,47]. However, our intriguing finding is that NaLaRuO$_4$, NaPrRuO$_4$ and NaNdRuO$_4$ RP oxides are also NCS with piezo-active symmetries. Thus, these compounds add to the growing list of NCS metals[19,20] or half-metals with unusual coexisting properties (broken inversion symmetry and metallic-like conduction).

In contrast, the NCS NaGdRuO$_4$ and NaYRuO$_4$ are found to be insulating with a gap appearing in both spin-up and spin-down electron channels (see Supplementary Fig. 11). We note that ruthenium oxides with antiferromagnetic insulating ground states are also not uncommon. For example, RP $Ca_2RuO_4$ is a

**Table 4 | The total energy difference and thermodynamic stability for different known and predicted RP phases from Quantum ESPRESSO[63].**

| RP oxides | Crystal symmetries from phonon calculations (ΔE) | | | | | | | Machine learning (ΔE) | | | ΔE$^D$ |
|---|---|---|---|---|---|---|---|---|---|---|---|
| | P4/nmm | Pmn2$_1$ | Pc | P$\bar{4}$2$_1$m | P$\bar{4}$2m | I$\bar{4}$2m | Pnma | P2$_1$2$_1$2 | Pbcm | Pca2$_1$ | |
| *Known composition* | | | | | | | | | | | |
| Ca$_2$IrO$_4$ (Pbca) | — | — | — | — | — | — | — | — | — | — | +34 |
| Ca$_2$IrO$_4$ (I4/mmm) | — | — | — | — | — | — | — | — | — | — | +156 |
| *New predictions* | | | | | | | | | | | |
| Stannates | | | | | | | | | | | |
| NaLaSnO$_4$ | 2.3 | 1.7 | 1.7 | 0 | 2.4 | 2.1 | 2.3 | 0 | 0.9 | 0.3 | +68.6 |
| NaPrSnO$_4$ | 9.5 | 9.3 | 9.3 | 0 | 9.5 | 3.4 | 9.5 | 0 | 3.4 | 2.9 | +79.9 |
| NaNdSnO$_4$ | 14.7 | 14.7 | 15.4 | 0 | 14.4 | 3.9 | 14.7 | 0 | 5.4 | 1.3 | +81.2 |
| NaGdSnO$_4$ | 40.2 | 34.8 | 34.5 | 0 | 28.0 | 5.4 | 35.2 | 0 | 14.6 | 10.9 | +75.6 |
| NaYSnO$_4$ | 46.8 | 37.1 | 36.4 | 0 | 32.5 | 5.9 | 37.6 | 0 | 16.6 | 11.7 | +73.6 |
| Ruthenates | | | | | | | | | | | |
| NaLaRuO$_4$ | 5.7 | 5.1 | 5.1 | 0 | 4.9 | 2.6 | 5.1 | 0 | 0.5 | 0.4 | +72.2 |
| NaPrRuO$_4$ | 15.5 | 14.9 | 14.9 | 0 | 10.9 | 4.6 | 14.9 | 0 | 1.8 | 0.7 | +78.3 |
| NaNdRuO$_4$ | 21.1 | 20.4 | 20.4 | 0 | 13.8 | 5.0 | 21.1 | 0 | 2.8 | 0.3 | +53.3 |
| NaGdRuO$_4$ | 46.1 | 41.7 | 41.7 | 1.0 | 26.6 | 7.1 | 43.2 | 1.03 | 8.9 | 0 | −14.1 |
| NaYRuO$_4$ | 179.9 | 47.6 | 47.6 | 2.6 | 32.7 | 8.8 | 49.2 | 2.6 | 11.4 | 0 | −1.3 |

DFT, density-functional theory; RP, Ruddlesden–Popper; OQMD, Open Quantum Materials Database.
The total energy difference ΔE (in units of meV per atom) is taken with respect to the lowest energy phase. Crystal symmetry with ΔE = 0 is identified as the ground state structure. For all ruthenates, we imposed ferromagnetic spin order on the Ru atom. ΔE$^D$ in meV per atom is the total energy difference calculated from DFT for a decomposition reaction obtained from OQMD[50,51]. Negative and positive values for ΔE$^D$ indicate that the compound is thermodynamically stable and unstable, respectively. Corresponding decomposition reactions are given in Supplementary Note 4. For Ca$_2$IrO$_4$, space groups Pbca and I4/mmm are the theoretical ground state and high-symmetry structures[15], respectively. Furthermore, in stannates structures initialized with Pnma symmetry converged to P2$_1$/m when $R =$ La, Pr or Nd. Similarly, in ruthenates Pc structure converged to P1 when $R =$ Pr, Gd or Y.

known antiferromagnetic insulator in the CS *Pbca* space group (Fig. 3e) at low temperatures[48,49]. Thus, we predict $NaGdRuO_4$ and $NaYRuO_4$ as potential multiferroics with polar symmetry, antiferromagnetic spin order and a bandgap. Are these stannates and ruthenates also thermodynamically stable? We address this question in the next section.

*Thermodynamic stability.* We use grand canonical linear programming[50] to determine the thermodynamic stability for the predicted RP stannates and ruthenates. The 'reservoir' of stable compounds present in the Open Quantum Materials Database[51] were chosen to describe the theoretical convex hull.

| Table 5 \| Bandgap ($E_g$ in eV) at the HSEsol level for each $NaRSnO_4$ compound from VASP[69,70] in the NCS $P\bar{4}2_1m$ space group. | |
| --- | --- |
| **Compound** | $E_g$ **(eV)** |
| $NaLaSnO_4$ | 4.35 |
| $NaPrSnO_4$ | 4.45 |
| $NaNdSnO_4$ | 4.42 |
| $NaGdSnO_4$ | 4.34 |
| $NaYSnO_4$ | 4.34 |

HSE, Heyd-Scuseria-Ernzerhof; NCS, Noncentrosymmetric; VASP, Vienna *ab initio* Simulation Package.

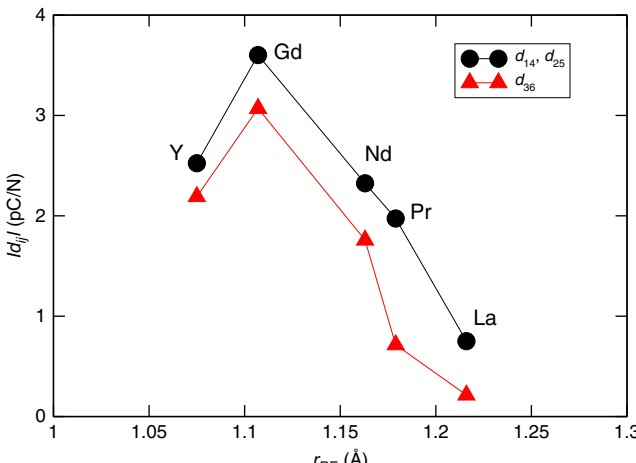

**Figure 5 | Calculated piezoelectric coefficients.** Piezoelectric strain coefficients (*y* axis) for the $P\bar{4}2_1m$ $NaRSnO_4$ structures as a function of the rare-earth cation ionic size in Å, $r_{RE}$ (*x* axis). There are three symmetry-allowed $d_{ij}$ components ($d_{14}$, $d_{25}$ and $d_{36}$) and two of which are equivalent ($d_{14} = d_{25}$).

The process involves calculation of the total energy change ($\Delta E^D$) for a chemical reaction involving reactants that are known to be thermodynamically stable and a product, which is the ground state structure of our predicted RP compounds. Compounds with negative $\Delta E^D$ are identified to be thermodynamically stable.

It is also important to note that compounds with positive $\Delta E^D$ (metastable) have also been synthesized. Commonly, when $\Delta E^D$ is $< +25$ meV per atom above the convex hull, it is suggested that the composition could be potentially synthesized under appropriate experimental conditions[52]. To evaluate this criterion for our design problem, we first calculated the $\Delta E^D$ for $Ca_2IrO_4$ that was recently epitaxially grown in the RP structure-type using the pulsed laser deposition method[29]. It is well known in the literature that $Ca_2IrO_4$ in RP structure type is a metastable phase[29]. Our main motivation is to compare the $\Delta E^D$ for $Ca_2IrO_4$ with our newly predicted compounds (especially those with positive $\Delta E^D$) and glean additional insights. The results are given in Table 4. The $\Delta E^D$ for RP $Ca_2IrO_4$ in the theoretical ground state and high-symmetry structures are $+34$ and $+156$ meV per atom, respectively, above the convex hull, yet it was successfully synthesized. We give the $\Delta E^D$ data for both the theoretical ground state and high-symmetry structures, because Souri *et al.*[29] do not report the crystal symmetry of their thin film, and therefore the reference point is unclear.

Having benchmarked the $\Delta E^D$ data for $Ca_2IrO_4$, we return to our predicted NCS stannates and ruthenates. In Table 4, we provide the $\Delta E^D$ data for both stannates and ruthenates. The associated decomposition reactions are given in the Supplementary Note 4. Two out of 10 compounds—$NaGdRuO_4$ and $NaYRuO_4$—have negative $\Delta E^D$, and therefore, we identify them to be thermodynamically stable and promising for synthesis. The remaining eight compounds have $\Delta E^D \leq +82$ meV per atom.

*Additional predictions.* In Table 7, we report our results for nine additional randomly chosen compounds that were predicted to have NCS ground state structures from ML. The total energy data, along with the different crystal symmetries obtained from both phonon calculations and ML, are given in the Supplementary Table 3. Seven out of nine compounds are found to have NCS ground state structures, in good agreement with our classification learning. Note that some of them (for example, $KBaNbO_4$ and $NaCaTaO_4$) have space groups that are not seen in any known or reported RP compounds (see Fig. 4). This is because we did not constrain our DFT calculations to only known structures or those from ML, but performed phonon calculations and full structure relaxations. The decomposition energies, $\Delta E^D$, for all nine compounds are also given in Table 7. Six out of nine predicted compounds have either a negative $\Delta E^D$ (thermodynamically stable) or $\Delta E^D \leq 34$ meV per atom (that is, stable relative to $Ca_2IrO_4$), indicating promise. Experimental

**Table 6 | Total energy difference ($\Delta E$ in meV per atom) with respect to the lowest energy structure for $NaRRuO_4$ in two $P\bar{4}2_1m$ and $Pca2_1$ structures with both FM and AFM spin configurations.**

| Compound | $\Delta E$ | | | | $\mu_B^{Ru}$ |
| --- | --- | --- | --- | --- | --- |
| | $P\bar{4}2_1m$ **FM** | $P\bar{4}2_1m$ **AFM** | $Pca2_1$ **FM** | $Pca2_1$ **AFM** | |
| $NaLaRuO_4$ | 0 | 7.3 | 0.4 | 6.0 | 0.91 |
| $NaPrRuO_4$ | 0 | 6.8 | 0.7 | 1.8 | 0.91 |
| $NaNdRuO_4$ | 0 | 6.7 | 0.3 | 0.5 | 0.91 |
| $NaGdRuO_4$ | 2.5 | 8.7 | 1.4 | 0 | 0.85 |
| $NaYRuO_4$ | 8.1 | 14.1 | 5.5 | 0 | 0.84 |

AFM, antiferromagnetic; FM, ferromagnetic.
All compounds initialized with AFM $P\bar{4}2_1m$ converged to AFM $P2_12_12$ structures indicating evidence of spin–lattice coupling. Constraining AFM configuration in $P\bar{4}2_1m$ structures (where we fixed the lattice constants to that of FM $P\bar{4}2_1m$) only resulted in total energies higher than that for AFM $P2_12_12$. Structures with $\Delta E = 0$ represent the ground state configuration. $\mu_B^{Ru}$ is the absolute value for the magnetic moment per Ru-site (in Bohr magnetons) in the corresponding ground state structures.

results are necessary to confirm these predictions. In Table 3, chemistries for all 242 predicted RP oxides that show potential for NCS structures are listed. The DFT optimized ground state crystallographic information files for all 19 compounds can be downloaded from ref. 53.

As a general observation, we note that the NCS $P\bar{4}2_1m$ space group that we predict for 13 out of 19 compositions from DFT is also one of the most commonly observed experimental ground states[16,17] (also see Fig. 4) for the $n = 1$ RP compounds.

## Discussion

We developed a computational strategy built on the foundations of applied group theory, ML and DFT to design NCS RP compounds. In terms of the novelty of our informatics approach, we note that the use of irreps as class labels for ML is new to materials science. Normally, space groups are utilized. The role of group theory in our framework was to transform the space groups into irreps. From using irreps as class labels for ML, we were able to reduce the complexity of our classification problem from 9 to 6 class labels. Even after reducing the complexity, we found that our data set suffered from class imbalance. To address this deficiency, we applied the SMOTE algorithm to generate synthetic data points and then constructed an ensemble of decision trees for irrep classification. Our decision trees identified 242 new compositions (from screening ∼3,200 compositions) that show potential for NCS ground state. We tested our prediction for 19 compositions using DFT, among which 17 were validated to have an NCS ground state structure. We thus find

good agreement between our informatics-based predictions and DFT ground state structures. One of the major design outcomes is the identification of two new multiferroics ($NaGdRuO_4$ and $NaYRuO_4$), which were also determined to be thermodynamically stable.

It is also important to recognize that not all our ML predictions agreed with the DFT calculations. For example, $KLaIrO_4$ and $BaLaGaO_4$ were predicted to be NCS but our frozen-phonon calculations and full structural relaxations from DFT indicate disagreement (Table 7). Moreover, the inconsistencies are found to be pronounced when both A/A′ cations have relatively large ionic sizes (for example, K, Ba or La). Our DFT calculations reveal that the presence of large A/A′ cations significantly reduces the amplitude of octahedral tilting, which we ascribe to the steric effects. Our ML models appear to incorrectly classify them as NCS.

There are several ways to reduce such misclassification errors and improve our ML prediction accuracies. We list some of them here: First, one of the most promising directions is to synthesize the predicted materials and determine the crystal structure for each compound, which will allow us to augment our data set with new data points and retrain our ML models. We anticipate our ML models to learn rapidly from these new data points and improve their prediction accuracy in subsequent iterations[32]. Second, our current ML models are based on five decision tree classifiers; one of the natural extensions would be to construct more than five bootstrapped samples and generate additional decision trees (or apply a random forest algorithm with hundreds of classifiers) that could, in principle, reduce the misclassification

**Table 7 | DFT aided validation for nine randomly selected RP oxides that were predicted to have an NCS ground state structure from ML.**

| RP oxides | DFT ground state | NCS ground state (in %) as predicted from ML | Predicted space groups from ML [irrep label] | | | | | $\Delta E^D$ |
|---|---|---|---|---|---|---|---|---|
| | | | Tree 1 | Tree 2 | Tree 3 | Tree 4 | Tree 5 | |
| $NaLaHfO_4$ | $P\bar{4}2_1m$ (NCS) | 100 | $Pca2_1$ $[X_2^+ \oplus X_3^+]$ | $P2_12_12$ $[X_3^+ (\eta_1,\eta_2)]$ | $P\bar{4}2_1m$ $[X_3^+ (\eta_1,\eta_1)]$ | $Pca2_1$ $[X_2^+ \oplus X_3^+]$ | $P2_12_12$ $[X_3^+ (\eta_1,\eta_2)]$ | −17.9 |
| $NaLaZrO_4$ | $P\bar{4}2_1m$ (NCS) | 80 | $Pca2_1$ $[X_2^+ \oplus X_3^+]$ | $P2_12_12$ $[X_3^+ (\eta_1,\eta_2)]$ | $Pbcm$ $[X_3^+ (0,\eta_1)]$ | $Pca2_1$ $[X_2^+ \oplus X_3^+]$ | $P2_12_12$ $[X_3^+ (\eta_1,\eta_2)]$ | −22.6 |
| $NaLaIrO_4$ (FM) | $P\bar{4}2_1m$ (NCS) | 100 | $Pca2_1$ $[X_2^+ \oplus X_3^+]$ | $P2_12_12$ $[X_3^+ (\eta_1,\eta_2)]$ | $P\bar{4}2_1m$ $[X_3^+ (\eta_1,\eta_1)]$ | $Pca2_1$ $[X_2^+ \oplus X_3^+]$ | $P2_12_12$ $[X_3^+ (\eta_1,\eta_2)]$ | +204.6 |
| $KLaIrO_4$ (FM) | $Pbcm$ (CS) | 80 | $Pca2_1$ $[X_2^+ \oplus X_3^+]$ | $P\bar{4}2_1m$ $[X_3^+ (\eta_1,\eta_1)]$ | $Pca2_1$ $[X_2^+ \oplus X_3^+]$ | $Pca2_1$ $[X_2^+ \oplus X_3^+]$ | $Ibca$ $[P_4]$ | +135.4 |
| $KBaNbO_4$ | $P2_1$ (NCS) | 100 | $Pca2_1$ $[X_2^+ \oplus X_3^+]$ | $P\bar{4}2_1m$ $[X_3^+ (\eta_1,\eta_1)]$ | $Pca2_1$ $[X_2^+ \oplus X_3^+]$ | $Pca2_1$ $[X_2^+ \oplus X_3^+]$ | $Pca2_1$ $[X_2^+ \oplus X_3^+]$ | −832 |
| $NaCaTaO_4$ | $Pca2_1$ (NCS) | 100 | $Pca2_1$ $[X_2^+ \oplus X_3^+]$ | $Pca2_1$ $[X_2^+ \oplus X_3^+]$ | $P\bar{4}2_1m$ $[X_3^+ (\eta_1,\eta_1)]$ | $Pca2_1$ $[X_2^+ \oplus X_3^+]$ | $P2_12_12$ $[X_3^+ (\eta_1,\eta_2)]$ | +15.9 |
| $SrLaInO_4$ | $P\bar{4}2_1m$ (NCS) | 100 | $Pca2_1$ $[X_2^+ \oplus X_3^+]$ | $P2_12_12$ $[X_3^+ (\eta_1,\eta_2)]$ | $P2_12_12$ $[X_3^+ (\eta_1,\eta_2)]$ | $P2_12_12$ $[X_3^+ (\eta_1,\eta_2)]$ | $P2_12_12$ $[X_3^+ (\eta_1,\eta_2)]$ | +38.9 |
| $SrYGaO_4$ | $P2_1$ (NCS) | 80 | $P2_12_12$ $[X_3^+ (\eta_1,\eta_2)]$ | $\phi$ | $P2_12_12$ $[X_3^+ (\eta_1,\eta_2)]$ | $P2_12_12$ $[X_3^+ (\eta_1,\eta_2)]$ | $Pca2_1$ $[X_2^+ \oplus X_3^+]$ | +26.4 |
| $BaLaGaO_4$ | $P4/nmm$ (CS) | 60 | $Pbcm$ $[X_3^+ (0,\eta_1)]$ | $P2_12_12$ $[X_3^+ (\eta_1,\eta_2)]$ | $Pbcm$ $[X_3^+ (0,\eta_1)]$ | $P2_12_12$ $[X_3^+ (\eta_1,\eta_2)]$ | $P2_12_12$ $[X_3^+ (\eta_1,\eta_2)]$ | −51.1 |

CS, centrosymmetric; DFT, density-functional theory; FM, ferromagnetic spin order imposed on the Ir-atom; ML, machine learning; NCS, noncentrosymmetric structures.
Note that in a vast majority of compounds the DFT energy difference between space groups $P2_12_12$ and $P\bar{4}2_1m$ is of the order of few tenths of meV per atom. Additional details are given in Supplementary Table 3 and Supplementary Note 4. For $KBaNbO_4$, the structure initialized with $Pca2_1$ symmetry converged to $P2_1$ in our DFT calculations. $\Delta E^D$ (in meV per atom) is the decomposition energy for a chemical reaction given in Supplementary Note 4. Negative and positive values for $\Delta E^D$ indicate that the compound is thermodynamically stable and unstable, respectively.

errors. Also, exploring kernel-based ML algorithms, such as support vector machines and semisupervised learning schemes represent alternative informatics-based avenues to gain confidence or reduce uncertainties in our predictions.

Furthermore, we demonstrated the use of the SMOTE algorithm for the first time in materials design problems; recently, a number of new algorithms[35] have been developed for addressing similar class-imbalance problems, which could also be explored. We note that class-imbalance problems are ubiquitous in materials design and remains an unchartered territory in materials informatics[54]. Finally, the choice of more robust features could also improve the prediction accuracies. Further computational efforts aimed at exhaustively evaluating the potential energy surface of related phases[55] or alternatively, data-driven approaches[56] involving inference models could further refine the predictions by addressing issues related to compound formability and order-disorder transitions.

Notwithstanding the limitations, our approach provides a rational framework for structure-based design of novel functional materials with implications beyond the layered RP oxides. For instance, our methodology can be extended to explore NCS structures in Dion–Jacobson, Aurivillius, Brownmillerite or any crystal family. In principle, our strategy could also guide the search for materials with intriguing functionalities such as ferroaxiality[57]. The key component to realize such predictions will be the database construction process and more importantly, the nature of available data (including features) would determine the type of questions that can be addressed. In terms of ML methods, off-the-shelf classification learning with class-imbalance algorithms (such as those demonstrated in this work) has the potential to provide insights necessary for guiding the accelerated search of new materials with targeted crystal symmetry or functionality. Advanced learning strategies (for example, semisupervised learning, algorithms beyond SMOTE and Bayesian methods) may be necessary, but the choice and its formulation will hinge critically on the available databases and/or prior domain knowledge.

## Methods

**Group theory.** The group theoretical analysis was performed using the ISO-TROPY[58] tool and electronic resources available from the Bilbao Crystallographic Server[59].

**Materials informatics.** We used the following inference and ML methods in this paper: PCA for data-dimensionality reduction and feature extraction[60], sampling techniques such as bootstrap method that constructs multiple data sets from our experimental data set via sampling with replacement, decision tree classification learning[61] for formulating QCSR design rules and SMOTE[34] to rectify the class-imbalance problem. We chose the decision tree classification learner for the following reasons[62]: (i) they are interpretable making the model transparent to domain experts; (ii) the splitting criteria (for example, Shannon entropy) serves to accomplish feature selection without the need for using any additional ML methods; (iii) they are scalable; and (iv) they have the capability to match the prediction accuracies of state-of-the-art ML methods. ML calculations were performed using RSTUDIO and WEKA. The decision tree algorithm as implemented in WEKA was used. The data set was constructed using the Waber–Cromer orbital radii as features.

The class-imbalance problem was rectified using the SMOTE algorithm. When there is class-imbalance, these ML models could ignore the less frequently observed class labels and group them with other class labels in the nearest-neighbor high-dimensional data space that occur more frequently. This is not desirable for this work, because the frequency of occurrence of the NCS space groups, to begin with, are already under-represented. The input to SMOTE is our data set and three additional parameters: (i) the under-represented or minority class label that we intend to oversample, (ii) the number of nearest neighbours ($k$) and (iii) the number of extra synthetic data samples (in %) to be created. The SMOTE algorithm functions as follows: it takes the difference between the feature vectors (that is, orbital radii) of the under-represented irreps and its $k$ nearest neighbours and multiplies the difference by a random number between 0 and 1 to create a new feature vector. This new feature vector is augmented to the original data set. As a result, the selection of a random data point is made along the line segment

(a simplified visual representation of the process based on our data set is given in Supplementary Fig. 1). We used PCA to ensure that SMOTE did not affect the manifold of our data set. We use the SMOTE algorithm as implemented in WEKA[37].

**Electronic structure calculations.** DFT calculations for all RP compounds were performed using the planewave pseudopotential code, Quantum ESPRESSO (QE)[63] to obtain the total energies. We used ultrasoft pseudopotentials[64] with the PBEsol exchange-correlation functional[65] taken from the PSlibrary[66]. A plane-wave cutoff of 60 Ry was used during the ionic and electronic relaxation steps. Electron correlations in Ru-$4d$ and Ir-$5d$ electrons were treated using the Hubbard-$U$ method within the Dudarev formalism[67]. Spin-polarized calculations with collinear ferromagnetic spin order were imposed on the Ru and Ir atoms. An effective Hubbard-$U$ of 1.5 eV was chosen in both cases. Frozen phonon calculations were performed using PHONOPY code[68] that uses the forces from QE as input for calculating the dynamical matrices and interatomic force constants. We employed a supercell of size $2 \times 2 \times 2$ with 112 atoms for the frozen phonon calculations.

All calculations to obtain bandgaps and piezoelectric coefficients for NaRSnO₄ were performed using DFT as implemented in the Vienna ab initio Simulation Package[69,70]. The crystal structures were taken from converged QE calculations. We used projector augmented-wave potentials[71] with the PBEsol functional. The piezoelectric and elastic tensors were computed within the density-functional perturbation theory[72,73] with a plane-wave cutoff of 800 eV. The density of states were computed first with PBEsol, and then with different amounts of exact exchange using HSE (Heyd–Scuseria–Ernzerhof). By comparing the experimental bandgap of BaSnO₃ with our computed values, we selected the amount of exact exchange to use (here 35%).

**Data availability.** The data sets for the informatics study and the DFT optimized crystallographic information files are deposited at figshare (refs 41,53.).

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

## Acknowledgements

P.V.B. and T.L. acknowledge funding support from the Los Alamos National Laboratory (LANL) LDRD no. 20140013DR on Materials Informatics and the Center

for Nonlinear Studies (CNLS). J.M.R. and J.Y. were supported by NSF under grant nos. DMR-1454688 and DMR-1420620, respectively. The authors acknowledge the High-Performance Computing Modernization of the DOD and LANL Institutional Computing (IC) for computational resources that have contributed to the research results reported herein.

## Author contributions

The study was planned, calculations performed and the manuscript prepared by P.V.B., J.Y., T.L. and J.M.R. All authors discussed the results, wrote and commented on the manuscript.

## Additional information

**Competing financial interests:** The authors declare no competing financial interests.

**Publisher's note**: 

