## [Peer Review File · Nature Communications]

Reviewers' comments:

Reviewer #1 (Remarks to the Author):

This is an interesting and important paper in developing the methodology for combining group theory, material informatics, machine learning and DFT to provide a large scale search for new noncentrosymmetric layered oxides. Nearly 250 new compositions are identified, which is quite impressive. The work builds on a prior foundation laid in reference 17 where group theoretical methods and Bayesian analysis was used to predict new 214 RP iridates.

As I understand, the main advancement in this current work over that prior work is the machine learning component, and the large number of new predictions presented here. This distinction from prior work could perhaps be more explicitly stated upfront.

Another suggestion I have is to discuss in greater detail how easy it is to expand this approach to searching other structure classes or phenomena. For example, instead of 214 RP, how about other RPs? How about Dion Jacobsen, Aurivillius, brown-millerites, etc. How do we begin to expand similar approaches to say discover new mechanisms and material classes for improper ferromagnetism or ferroaxiality?

Overall, I believe this is an excellent contribution to the field, and I highly recommend its publication. It will spur not only new theory methods in materials genomics, but also a large experimental activity worldwide.

Reviewer #2 (Remarks to the Author):

The targeted synthesis of new solid state materials that lack inversion symmetry is a long-standing challenge. The use of machine learning technique to screen potential candidate compounds could provide an important advancement in this field and could greatly increase the rate at which new NCS materials are discovered. Moreover, this work represents a step forward in the inclusion of solid state structure in machine learning studies.

The ability to get physically meaningful linear combinations using PCA is referenced, but the validation that this is the case in the presented work needs to be better supported.

The authors should state why j48 decision trees are the model of choice. The authors do mention that constructing more than five bootstrapped samples is a logical next step, in addition to random forests or support vector machines. However, accuracy metrics for different model choices are not presented.

Applying the informatics as a guide for the DFT calculations is a valid approach, but since the recommended compositions are not experimentally validated, a complete listing of the recommended compositions should be listed.

The selection of NaDyTiO₄, NaSmTiO₄ and NaHoTiO₄ as essentially the test set to validate the ability of their classification learning to correctly identify the labels correctly is important. These compounds are chemically and structurally similar, and each having the same irrep label. Selecting a more diverse test set might provide a more robust test of this approach.

Experimental testing of the identified compositions would, as noted by the authors, allow for further refinement and improvement of the model. While a synthetic component would strengthen this manuscript, it is not required for this paper.

In the first paragraph, the authors state that NCS materials break all mirror symmetry elements, improper rotations and centers of symmetry. Mirror symmetry elements are of course allowed in achiral NCS structures. In addition, a fourth subset of NCS structures are those that exhibit

circular dichroism (in addition to polar, piezoelectric and enantiomorphic).

Reviewer #3 (Remarks to the Author):

In this manuscript, the authors use machine learning methods to identify potential new noncentrosymmetric Ruddlesden-Popper (RP) materials. The work appears to be technically sound, but I do not believe it meets the criteria required for publication in Nature Communications.

The core method used by the authors, involving the use of orbital radii, principal component analysis, and a decision tree to classify crystal structures, has already been published by this same PI (Nature Scientific Reports 5, Article number 13285 (2015)). The authors have modified the method in this manuscript by using it to predict space groups and irreducible representations rather than crystal structure types, but this is a relatively minor change.

The main product of this project is the identification of 248 candidate compositions that the algorithm indicates are likely to result in noncentrosymmetric RP compounds. The authors used density functional theory (DFT) to compare the energies of possible centrosymmetric and noncentrosymmetric RP structures for eight of these compositions, and they found that in all eight cases the noncentrosymmetric phase had lower energy. Based on these results, the authors recommend these phases for experimental synthesis and characterization.

This paper would have been much stronger had the authors collaborated with an experimental group to demonstrate that their predictions could be experimentally realized. However they only attempted to validate their predictions using DFT, and as far as I can tell they only evaluated 8 of their 248 candidate compounds. It is not clear to me why they didn't simply run DFT calculations on the remaining 240. It would have been feasible with only moderate computational expense, and it would have provided a much more compelling demonstration of the validity of their approach. Even the results for the 8 compounds they evaluated appear to be only partial, as they only compared a few possible competing structures for these compounds. They do not provide a more complete assessment of the calculated thermodynamic stability of these compounds, which would seem to be an important step before recommending attempted synthesis. Assessing thermodynamic stability against a database of competing products could probably have been done readily through the use of online tools such as the Materials Project or the Open Quantum Materials Database.

As a lesser concern, the manuscript reads at times as overly technical with superfluous details. Some of the details are a little odd, such as the analysis to determine whether the principal components were orthogonal to each other. Principal components are always orthogonal to each other, by definition.

Overall, this manuscript would likely be of interest to some researchers in this field, but the method lacks significant novelty compared to the PI's previously published work and the results are not sufficiently developed to be of extreme importance or attract broad interest. Should the authors wish to continue to pursue publication in Nature Communications, I would suggest that they provide more compelling evidence that through their method they have discovered at least one new technologically significant material.

June 2, 2016

Letter to the Referees

Note: Our response to referees' comments are given in purple text color.

Report of the First Referee – NCOMMS-16-01058

Response to Referee #1

Comment: This is an interesting and important paper in developing the methodology for combining group theory, material informatics, machine learning and DFT to provide a large scale search for new noncentrosymmetric layered oxides. Nearly 250 new compositions are identified, which is quite impressive. The work builds on a prior foundation laid in reference 17 where group theoretical methods and Bayesian analysis was used to predict new 214 RP iridates.

As I understand, the main advancement in this current work over that prior work is the machine learning component, and the large number of new predictions presented here. This distinction from prior work could perhaps be more explicitly stated upfront.

Response: We thank the reviewer for his/her comments. In the revised manuscript, we have stated upfront the distinction between the current work and the earlier work. The revision on Page 2 (in Section: Results and Subsection: Approach) reads as follows:

“We note that this paper is a significant advancement from the earlier work of Balachandran *et al*¹⁹, where the emphasis was on enumerating symmetry guidelines.”

Comment: Another suggestion I have is to discuss in greater detail how easy it is to expand this approach to searching other structure classes or phenomena. For example, instead of 214 RP, how about other RPs? How about Dion Jacobsen, Aurivillius, brown-millerites, etc. How do we begin to expand similar approaches to say discover new mechanisms and material classes for improper ferromagnetism or ferroaxiality?

Response: This is an interesting question and an important one, because it focuses on the applicability of our strategy beyond $n = 1$ RP phases. We note that our methodology is very generic and can be extended to Dion-Jacobson, Aurivillius, Brownmillerites or any structure-type. The key component will be the database and the nature of available data would determine the type of questions that one can address. For example, if the objective is to identify new materials with ferroaxial transitions, then one of the first steps should be to build a database of known crystal chemistries and “label” those that show ferroaxial transitions and also those that do not show ferroaxial transitions. These labels form the “class label” for classification learning. Orbital radii can be used as features to describe each of those crystal chemistries uniquely in the database. Balachandran *et al* [Nature Scientific Reports 5, Article number 13285 (2015)] have shown that these features are transferable and available for all elements in the periodic table. Alternatively, other common crystal chemistry features (e.g. ionic radii, electronegativity) can also be employed. If the number of data points in each of the class label is well represented, then there is no class-imbalance problem and the database is ready for classification learning. In contrast, when one of the class labels is under-represented (similar to this work), then class-imbalance algorithms (e.g. SMOTE) become necessary prior to classification learning. Machine learning methods, such as principal component analysis (PCA) and decision tree algorithms are well known and readily implemented in many mathematical packages (e.g. Python, R and Matlab to name a few). The group theoretical treatment is also applicable for these crystal structure-types.

We have added a new paragraph in the **Discussion** section on Page 9, which reads as follows:

“Our approach provides a rational framework for structure-based design of functional materials with implications beyond the layered RP oxides. For instance, our methodology can be extended to explore NCS structures in Dion-Jacobson, Aurivillius, Brownmillerite or any crystal family. In principle, our strategy could also guide the search for materials with intriguing functionalities such as ferroaxiality⁹⁵. The key component to realize such predictions will be the database construction process and more importantly, the nature of available data (including features) would determine the type of questions that can be addressed. In terms of ML methods, off-the-shelf classification learning with class-imbalance algorithms (such as those demonstrated in this work) has the potential to provide insights necessary for guiding the accelerated search of new materials with targeted crystal symmetry or functionality. Advanced learning strategies (e.g. semi-supervised learning, algorithms beyond SMOTE and Bayesian methods) may be necessary, but the choice and its formulation will hinge critically on the available database.”

Comment: Overall, I believe this is an excellent contribution to the field, and I highly recommend its publication. It will spur not only new theory methods in materials genomics, but also a large experimental activity worldwide.

Response: We thank the reviewer for capturing the merits of our work and recommending it for publication.

Report of the Second Referee – NCOMMS-16-01058

Comment: The targeted synthesis of new solid state materials that lack inversion symmetry is a long-standing challenge. The use of machine learning technique to screen potential candidate compounds could provide an important advancement in this field and could greatly increase the rate at which new NCS materials are discovered. Moreover, this work represents a step forward in the inclusion of solid state structure in machine learning studies.

Response: We thank the reviewer for his/her comments.

Comment: The ability to get physically meaningful linear combinations using PCA is referenced, but the validation that this is the case in the presented work needs to be better supported.

Response: We apologize for the lack of clarity. We used PCA to reduce the dimensionality of the data from 22 to 8 column vectors, yet capturing > 90% of the variation in the data. Each PC is a linear combination of the weighted contribution of orbital radii, and we show all PC's in the Supplementary Figure 2. We now turn our attention to the decision tree shown in Supplementary Figure 4 and follow the path $PC1 \leq -2.6796$ AND $PC2 \leq -0.1335$ AND $PC5 \leq 0.152 \rightarrow X_3^+(\eta_1, \eta_2)$ in the leaf node. From Supplementary Figure 2, the following orbitals are identified as important for predicting the irrep $X_3^+(\eta_1, \eta_2)$ using our decision tree:

- PC1: Orbital radii A-5p, A-6s, A-4f, B-4s, B-3d, B-5s and B-4d are important, because their weighted contributions are relatively larger than that of other orbital radii.
- PC2: A-2p, A-3s, B-6s, B-5d, B-4p, B-5s and B-4d
- PC5: A-4p, A-5s, A-4d, A-5p, A-6s, B-4s, B-3d, B-5s and B-4d

Projected density of states (PDOS) from DFT calculations for RP compounds with $X_3^+(\eta_1, \eta_2)$ octahedral distortions in the ground state would allow us to validate this finding. Exploring changes in orbital bandwidths and shifts in their center-of-mass would permit us to glean insights necessary for describing the stability of a crystal structure (or distortions). Thus, one can potentially extract physical meaning from PCA and decision trees. We do not carry out the electronic structure calculations here, because we anticipate the decision trees to evolve as more compounds are validated and fed back for re-training our models.

We have added this discussion in the Supplementary Note 1.

Comment: The authors should state why j48 decision trees are the model of choice.

Response: We chose the decision tree classification learner for the following reasons: (i) They are interpretable making the model transparent to domain experts; (ii) The splitting criteria (e.g. Shannon entropy) serves to accomplish feature selection without the need for using any additional ML methods; (iii) They are scalable; and (iv) They have the capability to match the prediction accuracies of state-of-the-art ML methods. We discuss these points in the **Methods** section and **Materials Informatics** subsection in our original and the resubmitted manuscript.

Comment: The authors do mention that constructing more than five bootstrapped samples is a logical next step, in addition to random forests or support vector machines. However, accuracy metrics for different model choices are not presented.

Response: We did not present the accuracy metrics, because we did not perform classification learning using random forests or support vector machines. Those recommendations were intended for future work. In this paper, all conclusions were made from utilizing decision trees as the classifier. The reasons for considering decision trees are discussed in our response to the previous comment.

Comment: Applying the informatics as a guide for the DFT calculations is a valid approach, but since the recommended compositions are not experimentally validated, a complete listing of the recommended compositions should be listed.

Response: We have now added a new table (Table 4) on Page 8 in the revised manuscript, where all predicted NCS compositions are listed.

Comment: The selection of NaDyTiO₄, NaSmTiO₄ and NaHoTiO₄ as essentially the test set to validate the ability of their classification learning to correctly identify the labels correctly is important. These compounds are chemically and structurally similar, and each having the same irrep label. Selecting a more diverse test set might provide a more robust test of this approach. Experimental testing of the identified compositions would, as noted by the authors, allow for further refinement and improvement of the model. While a synthetic component would strengthen this manuscript, it is not required for this paper.

Response: This is a very good question and we thank the reviewer for bringing this point to our attention. One of the reasons for choosing those particular compounds was that they are experimentally known to be noncentrosymmetric (NCS) and therefore, we wanted to test whether our models could predict the structures for these chemistries first. But, the reviewer's point is very well taken and we agree that validation using diverse chemical compositions will further strengthen the model generalizability and robustness.

We have now tested our decision tree classifiers using a more diverse test set. Our results are given in a new table (Table 3 in the revised manuscript on Page 6) and discussed in the subsection **Classification learning** on Page 6. Our decision trees predict with $\geq 60\%$ accuracy, 12 out of 14 compositions in the independent test set giving confidence to our classification learning.

Comment: In the first paragraph, the authors state that NCS materials break all mirror symmetry elements, improper rotations and centers of symmetry. Mirror symmetry elements are of course allowed in achiral NCS structures. In addition, a fourth subset of NCS structures are those that exhibit circular dichroism (in addition to polar, piezoelectric and enantiomorphic).

Response: We thank the reviewer for bringing this to our attention. We have revised the first paragraph in the **Introduction** to accommodate the comments.

Report of the Third Referee – NCOMMS-16-01058

Response to referee #3

Comment: In this manuscript, the authors use machine learning methods to identify potential new non-centrosymmetric Ruddlesden-Popper (RP) materials. The work appears to be technically sound, but I do not believe it meets the criteria required for publication in Nature Communications.

The core method used by the authors, involving the use of orbital radii, principal component analysis, and a decision tree to classify crystal structures, has already been published by this same PI (Nature Scientific Reports 5, Article number 13285 (2015)). The authors have modified the method in this manuscript by using it to predict space groups and irreducible representations rather than crystal structure types, but this is a relatively minor change.

Response: We thank the reviewer for his/her comments. We understand the confusion from not clarifying the novelty of our approach compared to the paper published in the *Nature Scientific Reports 5, Article number 13285 (2015)*. The following points highlight some of the key innovations introduced in this paper:

1. The idea of using irreducible representations (irreps) as class labels for machine learning is new to materials science. Normally, space groups are utilized. Transforming space groups into irreps was the purpose of exploring and integrating group theoretical methods into our computational approach. Using irreps as class labels, we were able to reduce the complexity of our classification problem from 9 to 6 class labels. This is a key innovative idea, which was not developed in the *Nature Scientific Reports 5, 13285 (2015)* article.
2. Furthermore, our dataset also suffers from class-imbalance, where some of the class labels occur more frequently than the others. This is problematic for classification learning. To address this important deficiency, we introduced the SMOTE (Synthetic Minority Oversampling Technique) algorithm, which has not been used before in materials science or materials informatics literature (including the *Nature Scientific Reports* paper).
3. Another distinction is in the utilization of ensemble of decision trees (as opposed to using only one decision tree in the *Nature Scientific Reports* paper) for classification learning. Therefore, we believe that there is sufficient novelty in the approach developed in this paper as well as in the findings.

In the *Nature Scientific Reports* paper, we benchmarked our machine learning approach (namely combining principal component analysis and decision trees) on well established datasets. The benchmarking effort provided the confidence that our features and decision tree method can be extended to new datasets.

Comment: The main product of this project is the identification of 248 candidate compositions that the algorithm indicates are likely to result in noncentrosymmetric RP compounds. The authors used density functional theory (DFT) to compare the energies of possible centrosymmetric and noncentrosymmetric RP structures for eight of these compositions, and they found that in all eight cases the noncentrosymmetric phase had lower energy. Based on these results, the authors recommend these phases for experimental synthesis and characterization.

This paper would have been much stronger had the authors collaborated with an experimental group to demonstrate that their predictions could be experimentally realized. However they only attempted to validate their predictions using DFT, and as far as I can tell they only evaluated 8 of their 248 candidate compounds. It is not clear to me why they didn't simply run DFT calculations on the remaining 240. It would have been feasible with only moderate computational expense, and it would have provided a much more compelling demonstration of the validity of their approach. Even the results for the 8 compounds they evaluated appear to be only partial, as they only compared a few possible competing structures for these

compounds.

Response: Our DFT work involves performing frozen-phonon calculations on a $2 \times 2 \times 2$ supercell that contains 112 atoms. In the calculated phonon band structures, we anticipate phonons with negative (or imaginary) frequencies, which we then “freeze-in” and fully relax the resulting structure. Some of the low symmetry configurations require unit cell transformations (e.g. $\sqrt{2}a \times \sqrt{2}a \times c$ or $\sqrt{2}a \times \sqrt{2}a \times 2c$, where a and c are the unit-cell constants for the aristotype tetragonal $P4/nmm$ structure with 14 atoms) that doubles (28 atoms) or quadruples (56 atoms) the number of atoms in the unit cell, relative to the high symmetry aristotype structure. These are computationally intensive calculations, especially when we have transition metal atoms with correlated d -electrons (e.g. Ru or Ir). In Supplementary Table 1, we have reported the total energies for all structure obtained from “freezing-in” the unstable modes from the frozen-phonon calculations. Therefore, we firmly believe that we have explored the phase space and possible competing structures with sufficient rigor.

But, the point of the referee is well taken. Therefore, in the revised manuscript, we report results for an additional 11 new compositions. The results are given in Table 5 (on Page 8) and Table 7 (on Page 9) and discussed on Page 8 under the subsection **Additional Predictions**. Total energy data is provided in the Supplementary Table 1. We have chosen diverse B-site cations (e.g. In^{3+} , Ga^{3+} , Ir^{4+} , Hf^{4+} , Zr^{4+} , Nb^{5+} and Ta^{5+}) for our calculations to capture the overall chemical spread in our predicted NCS chemical space. We find that 16 out of 19 compositions have NCS ground state crystal structures validating our machine learning predictions. We agree that an experimental component would have significantly strengthened the impact of our work. Nevertheless, we are hopeful that our computational work would trigger several new experimental activities (also see Last Comment from Referee 1).

We also corrected the number of new predictions from 248 to 242 (six were doubly counted) in the revised manuscript.

Comment: They do not provide a more complete assessment of the calculated thermodynamic stability of these compounds, which would seem to be an important step before recommending attempted synthesis. Assessing thermodynamic stability against a database of competing products could probably have been done readily through the use of online tools such as the Materials Project or the Open Quantum Materials Database.

Response: Thermodynamical stability is an important data and we agree that such data can aid in experimental efforts. In the revised submission, we have calculated the formation enthalpy (ΔH corresponding to the chemical reaction $\text{Na}_2\text{O} + 2\text{BO}_2 + R_2\text{O}_3 \rightarrow 2\text{NaRBO}_4$, where $B=\text{Sn}$ or Ru and $R=\text{La}$, Pr , Nd , Gd or Y) for both NaRSnO_4 and NaRRuO_4 . The results are given in Table 5 (on Page 8) in the revised manuscript.

We find that all stannates have negative ΔH and all ruthenates (except the NaYRuO_4) have negative ΔH . Furthermore, assessing thermodynamic stability using well-established databases such as OQMD or Materials Project is difficult, because our calculations involve complex crystal structures (predicted for the first time and not found in ICSD repository) using PBEsol exchange-correlation functionals that are not reported in these databases.

Comment: As a lesser concern, the manuscript reads at times as overly technical with superfluous details. Some of the details are a little odd, such as the analysis to determine whether the principal components were orthogonal to each other. Principal components are always orthogonal to each other, by definition.

Response: We have revised our manuscript to remove the superfluous details.

Comment: Overall, this manuscript would likely be of interest to some researchers in this field, but the method lacks significant novelty compared to the PI's previously published work and the results are not sufficiently developed to be of extreme importance or attract broad interest. Should the authors wish to continue to pursue publication in Nature Communications, I would suggest that they provide more compelling evidence that through their method they have discovered at least one new technologically significant material.

Response: In our paper, we have discussed the ground state crystal structure and electronic functionalities of NaRSnO_4 , where $R=\text{La, Pr, Nd, Gd}$ and Y . We note that stannate crystal chemistries find application as potential transparent conducting oxides and our work predicts (for the first time) the band gaps and piezoelectric coefficients for these compounds in the $n = 1$ RP phase. We also showed that their piezoelectric coefficients can be controlled by the size of the rare-earth element and their piezoelectric properties are comparable to that of NaRTiO_4 . Our predictions of NCS RP stannates could potentially have technological impact in optoelectronic and sensor applications.

In our revised manuscript, on Pages 7–9 under the subsection **Ruthenates**, we have also discussed the ground state structures and electronic band structures for NaRRuO_4 , where $R=\text{La, Pr, Nd, Gd}$ and Y . Our DFT calculations show that all explored NaRRuO_4 ruthenates have NCS ground state structures, in good agreement with predictions from our classification learning. Our electronic structure calculations reveal that NaLaRuO_4 is metallic, whereas other ruthenates are half-metals (with a gap in the spin-up channel). We have provided the data in the Supplementary Figure 9. They, therefore, represent a unique class of noncentrosymmetric metals with unusual combination of properties. Recently, there is interest in studying noncentrosymmetric metals [see Kim *et al Nature* **533** pp. 68-72 (2016); Puggioni and Rondinelli, *Nature Communications* **5** 3432 (2015); Shi *et al Nature Materials* **12** pp. 1024-1027 (2013)]. Our predicted NCS NaRRuO_4 systems add to this intriguing, yet rare, new materials class, which we hope would trigger more theoretical and experimental activities. Furthermore, we also anticipate these ruthenates as candidates for exploring metal-insulator transitions with key implications in electronics applications.

Reviewers' comments:

Reviewer #1 (Remarks to the Author):

The response of the authors to the questions from the reviewers is thorough and well considered. There is significant additional work that is now included, including new classes of compounds, chemistries, and additional tables and subsections. This is an impressive treasure-trove of predictions for experimentalists, and its value towards furthering the field should be considered strongly. New classes of noncentrosymmetric correlated metals, piezoelectric metals (what would a stress do in such a metal?), and polar transparent conductors where optical response can perhaps be controlled by a field are sufficiently intriguing to interest a large number of experimentalists. Some of the literature already confirms their predictions, and that is an excellent start.

I like the author's clear response to referee R#3's first question about the novelty of this work in relation to the rest of the field. Unfortunately, this is not clearly captured in their changes to the manuscript in response to referee#1's first question on the same topic. I would recommend using the response to R#3 as the template for restating the novelty of this work.

Overall, I highly recommend this manuscript for publication!

Reviewer #2 (Remarks to the Author):

The authors have improved the quality of this manuscript through the first round of revisions. The new text about how this technique can be extended to different chemical systems and its reliance on accessible data is a welcome addition. Moreover, the introduction of structure into machine learning approaches is important and noteworthy.

A more explicit statement should be made demonstrating that ground states for the kind of materials discussed in this paper generally match well with observed experimental crystal structures. Particularly for a more general readership, it is not necessarily obvious that this should be the case. Such correlations do not always exist for even small molecule and protein crystallography.

It is good to note that in the Materials informatics portion, low temperature crystal symmetries were used, in line with their use of 0 K DFT calculations, and that crystal temperature dependence is understandably not included. A definition of "low temperature" should be included.

There should be some discussion as to why accuracy was used over balanced classification rate for determining the performance of the models. In addition, these results are somewhat over-reported in that the classification accuracy for the training, test and 10-fold methods are reported. I would suggest that in the light of the latter two, the accuracies for the training set self-predictions could be relegated to the supporting info (and possibly the 10-fold result also, since there is a properly withheld training set).

The bracketed numbers in Supplementary 4 - 8 should be defined in the figure captions.

Reviewer #3 (Remarks to the Author):

I thank the authors for their detailed responses to my comments. This manuscript has the potential to meet the criteria for publication in Nature Communications in two ways:

1. As an article on a novel, highly useful machine learning approach that is of broad interest to materials researchers.
2. As an article on the discovery of a new, important material (or group of materials).

Unfortunately I do not believe the article meets the criteria for publication in either of these ways. I would expect a Nature Communications article on machine learning to describe an approach that is highly novel, of broad interest, and rigorously evaluated. The current manuscript meets none of these criteria. The efficacy of their approach is evaluated only on a small, manually selected set of materials, rather than a randomized (or better yet, comprehensive) evaluation of their predictions. The authors correctly point out several ways in which their machine learning approach differs from their previous work, but these changes are largely incremental, natural extensions of what they have already done. I understand the argument that being the first to apply machine learning method X to material problem Y should merit publication in Nature Communications, but the literature is full of such articles in all sorts of journals.

I believe this manuscript has far more potential as an article on material discovery. If the authors are able to convincingly make the case that there is a high likelihood that they have discovered new, relevant, non-centrosymmetric materials, then I believe this article should be published. However the current version does not accomplish this. The authors would need to demonstrate that 1) for each composition the non-centrosymmetric structures are highly likely to be lower in energy than all other Ruddlesden-Popper-type structures and 2) there is a reasonable expectation that the materials they propose can be synthesized.

Regarding point 1), I appreciate the authors' point about the computational expense of a frozen-phonon calculation for each structure. However it is not clear to me how these frozen-phonon calculations were combined with the "space group theoretical analysis" to come up with the list of candidate structures at each composition. In Table 5, the same four candidate phases are listed for each of the compositions. However in Supplementary Table 1, there is a different set of structures listed for each composition. How were these structures derived? It would seem that for such a small validation set, comprehensively evaluating each of the compositions in each of the candidate structure types would not be very expensive, and it would allow for an apples-to-apples comparison across all compositions.

Regarding the assessment of synthesizability (point 2), I appreciate the authors' attempt to determine the thermodynamic stability of each of these structures. Unfortunately the approach they use is fundamentally flawed. It is not correct to declare that if the enthalpy of the reaction $\text{Na}_2\text{O} + 2\text{BO}_2 + \text{R}_2\text{O}_3 \rightarrow 2\text{NaRBO}_4$ is negative, then the material is "thermodynamically stable" (Page 7 of the revised manuscript). That would imply that the NaRBO_4 could only possibly decompose into $\text{Na}_2\text{O} + 2\text{BO}_2 + \text{R}_2\text{O}_3$, which of course is not true. The value of using the Materials Project or Open Quantum Materials Database (OQMD) is that through these tools, it is possible to identify the most likely decomposition products through the construction of convex hulls of known stable phases. For example, using the OQMD (<http://oqmd.org/analysis/gclp/>), it appears that a more likely decomposition pathway for NaLaRuO_4 is:

The enthalpy of this reaction would be a much better indicator of the thermodynamic stability of NaLaRuO_4 . The authors' argument that the OQMD and Materials Project data cannot be used because the databases are based on the PBE exchange-correlation functional is not convincing. Both of these projects have publicly documented how their calculations are done (see for example <http://oqmd.org/documentation/vasp>) and it would be straightforward to re-do the appropriate subset of the calculations in this paper in a way that is consistent with these databases. An alternative approach would be to simply use these databases to identify the likely decomposition products (as was done above) and directly calculate the energies of the proposed new NCS phases and possible decomposition products in any way the authors choose.

To get a better understanding of how to assess the thermodynamic stability of the proposed materials, I highly recommend the authors read the paper "Stability and electronic properties of new inorganic perovskites from high-throughput ab initio calculations" that was recently published by Korbelt et al. in *J. Mater. Chem. C*. In this paper, over 32,000 perovskite-type materials are screened and both OQMD and Materials Project data are used to assess the thermodynamic stability of newly discovered phases. Of particular note is that over 90% of the experimentally-known perovskites in the author's dataset had calculated decomposition energies of less than 25 meV / atom. This indicates that 25 meV / atom might be a reasonable cutoff for the authors of the current manuscript to use when identifying structures that are likely to be synthesizable.

Although it may seem that properly assessing thermodynamic stability before proposing new materials for synthesis is an unnecessary expense, it is not. It is easy - even trivial - to develop a new material in silico that has never been seen before and has remarkable properties. In the significant majority of these cases, the reason the material has never been seen before is because it is highly unstable. Without a proper assessment of thermodynamic stability, there is a risk that the manuscript in its current form could spur "large experimental activity worldwide" (as mentioned by Reviewer #1) that largely consists of chemists wasting time in a misguided effort to synthesize highly unstable materials.

September 15, 2016

Letter to the Referees

Note: Our response to referees' comments are given in purple text color.

Report of the First Referee – NCOMMS-16-01058A

Response to Referee #1

Comment: The response of the authors to the questions from the reviewers is thorough and well considered. There is significant additional work that is now included, including new classes of compounds, chemistries, and additional tables and subsections. This is an impressive treasure-trove of predictions for experimentalists, and its value towards furthering the field should be considered strongly. New classes of noncentrosymmetric correlated metals, piezoelectric metals (what would a stress do in such a metal?), and polar transparent conductors where optical response can perhaps be controlled by a field are sufficiently intriguing to interest a large number of experimentalists. Some of the literature already confirms their predictions, and that is an excellent start.

Response: We thank the reviewer for his/her comments on our paper.

Comment: I like the author's clear response to referee R#3's first question about the novelty of this work in relation to the rest of the field. Unfortunately, this is not clearly captured in their changes to the manuscript in response to referee #1's first question on the same topic. I would recommend using the response to R#3 as the template for restating the novelty of this work. Overall, I highly recommend this manuscript for publication!

Response: We thank the reviewer for recommending our paper for publication. We have now added a new paragraph on Page 10 under section **Discussions** that captures the novelty of our work, which reads as follows:

“We developed a computational strategy built on the foundations of applied group theory, machine learning and DFT to design NCS RP compounds. In terms of the novelty of our informatics approach, we note that the idea of using irreps as class labels for machine learning is new to materials science. Normally, space groups are utilized. The role of group theory in our framework was to transform the space groups into irreps. From using irreps as class labels for ML, we were able to reduce the complexity of our classification problem from 9 to 6 class labels. Even after reducing the complexity, we found that our dataset suffered from class-imbalance. To address this deficiency, we applied the SMOTE algorithm to generate synthetic data points and then constructed an ensemble of decision trees for irrep classification. Our decision trees identified 242 new compositions (from screening ~3,200 compositions) that show potential for NCS ground state. We tested our prediction for 19 compositions using DFT, among which 17 were validated to have an NCS ground state structure. We thus find good agreement between our informatics-based predictions and DFT ground state structures. One of the major design outcomes is the identification of two new multiferroics (NaGdRuO₄ and NaYRuO₄), which were also determined to be thermodynamically stable.”

Report of the Second Referee – NCOMMS-16-01058A

Comment: The authors have improved the quality of this manuscript through the first round of revisions. The new text about how this technique can be extended to different chemical systems and its reliance on accessible data is a welcome addition. Moreover, the introduction of structure into machine learning approaches is important and noteworthy.

Response: We thank the reviewer for his/her comments.

Comment: A more explicit statement should be made demonstrating that ground states for the kind of materials discussed in this paper generally match well with observed experimental crystal structures. Particularly for a more general readership, it is not necessarily obvious that this should be the case. Such correlations do not always exist for even small molecule and protein crystallography.

Response: It has been shown in the literature [see for example, Akamatsu *et al Physical Review Letters* **112** 187602 (2014); Mulder *et al Advanced Functional Materials* **23** 4810 (2013)] that the ground states predicted from density-functional theory (DFT) agree quite well with that of the experimental measurements for these RP compounds. We find that in the vast majority of our predicted compounds (15 out of 19), the ground state from DFT is the NCS $P\bar{4}2_1m$ space group. This is also the predominant experimentally observed NCS space group for the $n=1$ NCS RP titanates [Akamatsu *et al Physical Review Letters* **112** 187602 (2014); Gupta *et al Advanced Electronic Materials* **2** 1500196 (2016)]. We have included new text on Page 10 in the revised manuscript, which reads as follows:

“As a general observation, we note that the NCS $P\bar{4}2_1m$ space group that we predict for 13 out of 19 compositions from DFT is also one of the most commonly observed experimental ground states^{20,21} (also see Fig. 4) for the NCS $n=1$ RP compounds.”

Comment: It is good to note that in the Materials informatics portion, low temperature crystal symmetries were used, in line with their use of 0 K DFT calculations, and that crystal temperature dependence is understandably not included. A definition of “low temperature” should be included.

Response: Our definition of low temperature includes experimentally observed structures that are ≤ 300 K. Some RP compounds also undergo structural transformation at lower temperatures. Under such circumstance, we take the lower temperature crystal structure to be our label. We have incorporated these details on Page 3 (section Materials Informatics) in the revised manuscript, which now reads as follows.

“Our definition of low temperature includes experimentally observed structures for ≤ 300 K. Some RP compounds also undergo structural transformation at a much lower temperature (e.g. La_2NiO_4 ⁵³). Under such circumstances, we take the lower temperature crystal structure to be our label for informatics.”

Comment: There should be some discussion as to why accuracy was used over balanced classification rate for determining the performance of the models. In addition, these results are somewhat over-reported in

that the classification accuracy for the training, test and 10-fold methods are reported. I would suggest that in the light of the latter two, the accuracies for the training set self-predictions could be relegated to the supporting info (and possibly the 10-fold result also, since there is a properly withheld training set).

Response: We thank the reviewer for the suggestions. We provided our classification accuracy data over the balanced classification rate to give an account of the average performance of the decision tree models. We have now given the confusion matrix in the Supplementary Note 2 that shows class specific performance. In addition, we have also put the in-sample and 10-fold cross validation results in the Supplementary Information document (Supplementary Table 1). The average accuracies reported in the main manuscript are now defined based on how well the ensemble of decision trees performed on the out-of-sample withheld compounds.

Comment: The bracketed numbers in Supplementary 4 - 8 should be defined in the figure captions.

Response: The bracketed numbers at each leaf node correspond to the total number of RP compositions that reach the leaf. Sometimes we also find two numbers [e.g (7.0/1.0) as seen in Supplementary Figure 4]. The first number (7.0) is then the total number of compositions reaching the leaf node and the second number (1.0) is the number of misclassified compositions reaching the same leaf node. We have added these texts in the caption of each of the Supplementary Figures 4–8.

Report of the Third Referee – NCOMMS-16-01058A

Response to referee #3

Comment: I thank the authors for their detailed responses to my comments. This manuscript has the potential to meet the criteria for publication in Nature Communications in two ways:

1. As an article on a novel, highly useful machine learning approach that is of broad interest to materials researchers. 2. As an article on the discovery of a new, important material (or group of materials).

Unfortunately I do not believe the article meets the criteria for publication in either of these ways. I would expect a Nature Communications article on machine learning to describe an approach that is highly novel, of broad interest, and rigorously evaluated. The current manuscript meets none of these criteria.

The efficacy of their approach is evaluated only on a small, manually selected set of materials, rather than a randomized (or better yet, comprehensive) evaluation of their predictions. The authors correctly point out several ways in which their machine learning approach differs from their previous work, but these changes are largely incremental, natural extensions of what they have already done. I understand the argument that being the first to apply machine learning method X to material problem Y should merit publication in Nature Communications, but the literature is full of such articles in all sorts of journals.

Response: We thank the reviewer for his/her comments and we use this response to further clarify the novelty of our work. In this paper, we have carefully integrated methods borrowed from the machine learning literature (e.g. class-imbalance, dimensionality reduction, classification learning) with group theoretical tools and density-functional theory to accelerate the design and discovery of new functional materials. Our workflow (shown in Figure 2 in our main manuscript) is unique and we demonstrate it for the first time in materials science by designing new functional materials. One of the key outcomes from this work is the prediction of 242 new chemical compositions that show potential for noncentrosymmetric ground state structures, which is a ~ 25 -fold increase from what was known in the literature. We also validated the ground state structure for 19 of those compositions and found good agreement with machine learning. Our choice of stannates and ruthenates for DFT validation were motivated by the intriguing physics that they offer in RP crystal structure with implications towards technological applications.

We have now rigorously evaluated the energetics for the Ruddlesden-Popper (RP) stannates and ruthenates to determine their ground state. We now report a more thorough exploration of the total energy data by considering nine unique distorted crystal symmetries:

1. Six obtained from frozen-phonon calculations.
2. Three from machine learning (ML).

This new data is given in Table 4 on Page 8 of the revised manuscript. In addition, we have also evaluated the total energies for the five $\text{Na}R\text{RuO}_4$ compounds, where $R=\text{La, Pr, Nd, Gd}$ and Y , in an anti-ferromagnetic spin configurations for the top two lowest energy structures, namely $P\bar{4}2_1m$ and $Pca2_1$. This data is given in Table 6 on Page 9. Main outcomes from the rigorous total energy evaluation are the following: (i) NaLaRuO_4 is a ferromagnetic metal with piezo-active symmetry ($P\bar{4}2_1m$). (ii) NaPrRuO_4 and NaNdRuO_4 are ferromagnetic half-metals with piezo-active symmetry ($P\bar{4}2_1m$) and (iii) NaGdRuO_4 and NaYRuO_4 are anti-ferromagnetic insulators with polar symmetry ($Pca2_1$). The electronic band structure data are given in Supplementary Figure 11. Thus, we have predicted three new potential noncentrosymmetric (NCS) metals or half-metals and two new multiferroics. In addition, we have also predicted five new $\text{Na}R\text{SnO}_4$ with wide band gap and piezo-active crystal symmetries ($P\bar{4}2_1m$). We note that our revised manuscript provides sufficient data suggesting that the predicted compositions have intriguing functional properties and have the potential of being of wide interest to the condensed matter, materials science and solid-state chemistry communities.

Furthermore, we also validated our predictions for nine additional compositions. These compositions were chosen randomly, under one constraint that they should contain at least one B-site cation that show potential for NCS ground state as recommended by our ML (we discuss this in the last paragraph on Page 6). We

agree that comprehensive evaluation would have been ideal, but they are beyond our computational budget. Accelerating new materials discovery is a grand challenge problem and new approaches that complement existing strategies are desired to accomplish these objectives. Our work is an important step towards achieving these goals.

There are two novel components associated with materials informatics that have been explored for the first time in this paper: (i) Use of crystal symmetry representation in the form of irreducible representations (irreps) from group theory and mode crystallography (which Referee #2 in comment #1 notes as follows, “Moreover, the introduction of structure into machine learning approaches is important and noteworthy.”) (ii) Addressing class-imbalance problems using synthetic minority oversampling technique (SMOTE) algorithm that has not been utilized before in the materials informatics literature. Without irreps and SMOTE, we would not have predicted these materials in our first iteration. We agree with the reviewer that there are some natural extensions from the earlier Nature Scientific Reports 5, 13285 (2015) paper, such as extending our machine learning from binary to multi-class classification learning and moving from single classification model to an ensemble of classification models.

Comment: I believe this manuscript has far more potential as an article on material discovery. If the authors are able to convincingly make the case that there is a high likelihood that they have discovered new, relevant, non-centrosymmetric materials, then I believe this article should be published. However the current version does not accomplish this. The authors would need to demonstrate that 1) for each composition the non-centrosymmetric structures are highly likely to be lower in energy than all other Ruddlesden-Popper-type structures and 2) there is a reasonable expectation that the materials they propose can be synthesized.

Response: We thank the reviewer for the constructive comments and we have made revisions that address both bullet points (1) and (2). We summarize our main findings in this response and redirect the reviewer to Pages 7–9 in the revised manuscript for detailed discussions.

Point (1): for each composition the non-centrosymmetric structures are highly likely to be lower in energy than all other Ruddlesden-Popper-type structures. To address this point, we have expanded Table 4 on Page 9 (also reproduced as Table 1 in this Response Letter on Page 8 for referee’s convenience). For the stannates and ruthenates, (as noted in our response to previous comment) we identified a common set of six distorted structures from “freezing-in” the atomic displacements that correspond to the imaginary phonon modes in the phonon band structures (given in Supplementary Figures 9 and 10) and in addition, we also considered three more distorted structures ($P2_12_12$, $Pbcm$ and $Pca2_1$) as suggested by ML. This is how we combine the results from phonon calculations with our ML predictions. Therefore, for NaRSnO_4 and NaRRuO_4 we considered nine unique distorted RP structures to determine the ground state: $Pmn2_1$, Pc , $P\bar{4}2_1m$, $P42m$, $I\bar{4}2m$, $Pnma$, $P2_12_12$, $Pbcm$ and $Pca2_1$. We then fully relaxed the atomic coordinates and lattice geometry for these structures to obtain the total energy. The lowest energy structure is taken as the ground state structure. In the case of ruthenates, we also performed additional calculations where we evaluated the total energies for the five NaRRuO_4 compounds in one of the anti-ferromagnetic spin configurations for the top two lowest energy structures, $P\bar{4}2_1m$ and $Pca2_1$. Thus, we note that we have rigorously explored a large set of RP structures to determine the ground state.

Point (2): there is a reasonable expectation that the materials they propose can be synthesized. We have performed thermodynamic stability analysis using the OQMD website (as recommended by the reviewer) to determine the stability of our predicted compositions [see last column (ΔE^D) in Table 4 on Page 8 in the revised manuscript]. In the ruthenates, we find two (out of five) to be thermodynamically stable, which we consider to be promising for synthesis. The remaining eight compounds were found to have a positive decomposition energy, indicating metastability. To gain further insights, we also calculated the decomposition energy for Ca_2IrO_4 in the RP structure. Recently, Souril *et al* [in Nature Scientific Reports 6 25967 (2016)],

successfully synthesized the metastable Ca_2IrO_4 in the RP phase using Pulsed Laser Deposition. Our calculated decomposition energies for the theoretical ground state and high symmetry structures for Ca_2IrO_4 are +34 and +156 meV/atom, respectively. We use this data to benchmark our predictions, especially for our proposed compounds that have positive decomposition energies. We find that the decomposition energies for our proposed compounds fall within the bounds obtained for the Ca_2IrO_4 , which may serve as a guide for the experimentalists. Therefore, we are cautiously optimistic that these eight compounds could also be synthesized under appropriate synthesis conditions. We have revised the section on **Thermodynamic Stability** on Page 9 to address this question. The decomposition pathways from OQMD are given in Supplementary Note 4.

Comment: Regarding point 1), I appreciate the authors' point about the computational expense of a frozen-phonon calculation for each structure. However it is not clear to me how these frozen-phonon calculations were combined with the "space group theoretical analysis" to come up with the list of candidate structures at each composition. In Table 5, the same four candidate phases are listed for each of the compositions. However in Supplementary Table 1, there is a different set of structures listed for each composition. How were these structures derived? It would seem that for such a small validation set, comprehensively evaluating each of the compositions in each of the candidate structure types would not be very expensive, and it would allow for an apples-to-apples comparison across all compositions.

Response: This is an important point and we apologize for the lack of clarity. We redirect the reviewer to our previous two responses.

We also note that in our effort to determine the ground state structures for the additional nine predicted compositions, we followed the same strategy – consider distorted crystal symmetries from both phonon calculations and recommendations from machine learning. The complete list of crystal symmetries are given in the Supplementary Table 3. We understand that in some circumstances (e.g. NaLaZrO_4) our DFT validation set may represent only fewer distorted crystal symmetries. We note that these distorted symmetries are a direct reflection of our phonon calculations that uses a $2 \times 2 \times 2$ supercell with 112 atoms and we assume that is sufficient for determining the ground state.

Comment: Regarding the assessment of synthesizability (point 2), I appreciate the authors' attempt to determine the thermodynamic stability of each of these structures. Unfortunately the approach they use is fundamentally flawed. It is not correct to declare that if the enthalpy of the reaction $\text{Na}_2\text{O} + 2\text{BO}_2 + \text{R}_2\text{O}_3 \rightarrow 2\text{NaRBO}_4$ is negative, then the material is "thermodynamically stable" (Page 7 of the revised manuscript). That would imply that the NaRBO_4 could only possibly decompose into $\text{Na}_2\text{O} + 2\text{BO}_2 + \text{R}_2\text{O}_3$, which of course is not true. The value of using the Materials Project or Open Quantum Materials Database (OQMD) is that through these tools, it is possible to identify the most likely decomposition products through the construction of convex hulls of known stable phases. For example, using the OQMD (<http://oqmd.org/analysis/gclp>), it appears that a more likely decomposition pathway for NaLaRuO_4 is:
 $\text{NaLaRuO}_4 \rightarrow 0.125 \text{RuO}_2 + 0.25 \text{Na}_2\text{RuO}_3 + 0.125 \text{Ru} + 0.5 \text{NaLa}_2\text{RuO}_6$

The enthalpy of this reaction would be a much better indicator of the thermodynamic stability of NaLaRuO_4 . The authors' argument that the OQMD and Materials Project data cannot be used because the databases are based on the PBE exchange-correlation functional is not convincing. Both of these projects have publicly documented how their calculations are done (see for example <http://oqmd.org/documentation/vasp>) and it would be straightforward to re-do the appropriate subset of the calculations in this paper in a way that is consistent with these databases. An alternative approach would be to simply use these databases to identify the likely decomposition products (as was done above) and directly calculate the energies of the proposed new NCS phases and possible decomposition products in any way the authors choose.

To get a better understanding of how to assess the thermodynamic stability of the proposed materials, I highly recommend the authors read the paper "Stability and electronic properties of new inorganic perovskites from

high-throughput ab initio calculations” that was recently published by Korbelt et al. in J. Mater. Chem. C. In this paper, over 32,000 perovskite-type materials are screened and both OQMD and Materials Project data are used to assess the thermodynamic stability of newly discovered phases. Of particular note is that over 90% of the experimentally-known perovskites in the author’s dataset had calculated decomposition energies of less than 25 meV / atom. This indicates that 25 meV / atom might be a reasonable cutoff for the authors of the current manuscript to use when identifying structures that are likely to be synthesizable.

Although it may seem that properly assessing thermodynamic stability before proposing new materials for synthesis is an unnecessary expense, it is not. It is easy - even trivial - to develop a new material in silico that has never been seen before and has remarkable properties. In the significant majority of these cases, the reason the material has never been seen before is because it is highly unstable. Without a proper assessment of thermodynamic stability, there is a risk that the manuscript in its current form could spur “large experimental activity worldwide” (as mentioned by Reviewer #1) that largely consists of chemists wasting time in a misguided effort to synthesize highly unstable materials.

Response: We thank the reviewer for his/her constructive comments and appreciate referring to the work of Korbelt et al, OQMD (GCLP) and Materials Project for the decomposition pathway analysis. We completely agree with the assessment that thermodynamic analysis is critical to reliably inform the experimental community. We have read the suggested literature and thoroughly revised our thermodynamic stability analysis. We used the GCLP algorithm as implemented in OQMD and updated the decomposition reaction pathways. The new results are discussed on Pages 7–9. All decomposition pathways are given separately in Supplementary Note 4.

Table 1: The total energy difference and thermodynamic stability for different known and predicted RP phases from QUANTUM ESPRESSO⁸³. The total energy difference ΔE (in units of meV/f.u.) is taken with respect to the lowest energy phase. Crystal symmetry with $\Delta E=0$ is identified as the ground state structure. For all ruthenates, we imposed ferromagnetic spin order on the Ru-atom. ΔE^D in meV/atom is the total energy difference calculated from DFT for a decomposition reaction obtained from OQMD^{84,85}. Negative and positive values for ΔE^D indicates that the compound is thermodynamically stable and unstable, respectively. Corresponding decomposition reactions are given in the Supplementary Note 4. For Ca_2IrO_4 , space groups $Pbca$ and $I4/mmm$ are the theoretical ground state and high-symmetry structures¹⁸, respectively. Furthermore, in stannates structures initialized with $Pnma$ symmetry converged to $P2_1/m$ when $R=$ La, Pr or Nd. Similarly, in ruthenates Pc structure converged to $P1$ when $R=$ Pr, Gd or Y.

RP oxides	Crystal symmetries from Phonon calculations (ΔE)							Machine learning (ΔE)			ΔE^D
	$P4/nmm$	$Pmn2_1$	Pc	$P42_1m$	$P42m$	$I42m$	$Pnma$	$P2_12_12$	$Pbcm$	$Pca2_1$	
Known composition											
Ca_2IrO_4 ($Pbca$)	-	-	-	-	-	-	-	-	-	-	+34
Ca_2IrO_4 ($I4/mmm$)	-	-	-	-	-	-	-	-	-	-	+156
New predictions											
Stannates											
NaLaSnO_4	16	11.9	11.9	0	16.6	14.9	16	0	6.6	2.2	+68.6
NaPrSnO_4	66.6	65.4	65.5	0	66.9	23.5	66.5	0	24	20	+79.9
NaNdSnO_4	103.2	102.7	107.5	0	100.9	27.6	103.2	0	37.7	9.3	+81.2
NaGdSnO_4	281.6	244	241.6	0	196	37.9	246.6	0	102.3	76.3	+75.6
NaYSnO_4	327.6	260	255	0	227.3	41.7	263.2	0	116.5	82	+73.6
Ruthenates											
NaLaRuO_4	39.9	35.6	35.6	0	34.2	18.5	35.8	0	3.5	2.9	+72.2
NaPrRuO_4	108.5	104.4	104.4	0	76.4	32	105	0	12.3	5.2	+78.3
NaNdRuO_4	148	143	143.1	0	96.8	35.2	148	0	19.8	2.3	+53.3
NaGdRuO_4	322.4	291.7	291.8	7.2	186.3	49.4	302.6	7.2	62	0	-14.1
NaYRuO_4	1259	333.2	333.2	18.1	229.1	61.4	344.7	18.1	79.7	0	-1.3

Reviewers' comments:

Reviewer #3 (Remarks to the Author):

The authors have significantly improved the manuscript and addressed many of my concerns. In particular, Table 4 and the accompanying explanation are helpful. However it is potentially confusing to use two different units for the values in Table 4: meV / formula unit for $\Delta(E)$, and meV / atom for $\Delta(ED)$. I would suggest using meV / atom for all energy differences, which is a standard way of assessing relative stability between structures.

The authors have greatly improved their discussion of calculated thermodynamic stabilities, but this section of the manuscript could still use work. The authors present a list of nine randomly chosen structures in supplementary table 3, and state that "experimental results are necessary" to confirm their predictions for these compounds. That's not necessarily true, as these structures may be highly unstable, which would explain why some of those space groups have never been seen before in a RP compound. Calculated decomposition energies should be provided for these structures, as they are for the structures in Table 4.

The authors calculate the energies for two different structures of Ca_2IrO_4 , finding values that are 34 and 156 meV / atom above the convex hull. Then then state:

"We conjecture that our calculated $\Delta(ED)$ data for Ca_2IrO_4 could serve as an approximate working range for the experimentalists."

and later claim that:

"The remaining eight compounds have $\Delta(ED) +82$ meV/atom and fall within the bounds determined for the metastable Ca_2IrO_4 (discussed earlier). Therefore, we are cautiously optimistic that these eight compounds could also be synthesized under appropriate synthesis conditions, despite their positive $\Delta(ED)$ values."

Their conjecture is wrong. The calculated values for Ca_2IrO_4 do not form a range between which the decomposition energies of synthesizable compounds fall. Instead, the two different values for very similar structures indicate that one of the structures (the one with an energy that is 122 meV / atom higher) is not the experimentally observed state. An alternative explanation would be that the margin of error for their calculated stabilities is on the order of 120 meV / atom, but that would invalidate the rest of the claims made in the manuscript.

Based on their calculations, a reasonable range for synthesizable structures would be those with decomposition energies of less than 34 meV / atom, not those with decomposition energies between 34 and 156 meV / atom. This leaves NaGdRuO_4 and NaYRuO_4 as predicted NCS structures that have a reasonable potential to be synthesizable. I suggest the authors revise their manuscript to more realistically present their results.

October 25, 2016

Letter to the Referees

Note: Our response to referees' comments are given in purple text color.

Report of the Third Referee – NCOMMS-16-01058B

Response to Referee #3

Comment: The authors have significantly improved the manuscript and addressed many of my concerns. In particular, Table 4 and the accompanying explanation are helpful. However it is potentially confusing to use two different units for the values in Table 4: meV / formula unit for ΔE , and meV / atom for ΔE^D . I would suggest using meV / atom for all energy differences, which is a standard way of assessing relative stability between structures.

Response: We thank the reviewer for his/her comments. We have revised all total energy units to meV/atom throughout the paper.

Comment: The authors have greatly improved their discussion of calculated thermodynamic stabilities, but this section of the manuscript could still use work. The authors present a list of nine randomly chosen structures in supplementary table 3, and state that “experimental results are necessary” to confirm their predictions for these compounds. That’s not necessarily true, as these structures may be highly unstable, which would explain why some of those space groups have never been seen before in a RP compound. Calculated decomposition energies should be provided for these structures, as they are for the structures in Table 4.

Response: We have performed additional DFT calculations for those nine compositions to determine the thermodynamic stability. We have now updated Table 7 with the decomposition energies (ΔE^D) data. The decomposition reactions are also given in the Supplementary Note 4. The newly added texts on Page 10 now reads as follows,

“The decomposition energies, ΔE^D , for all nine compounds are also given in Table 7. Six out of nine predicted compounds have either a negative ΔE^D (thermodynamically stable) or $\Delta E^D \leq 34$ meV/atom (*i.e.* stable relative to Ca_2IrO_4), indicating promise.”

Comment: The authors calculate the energies for two different structures of Ca_2IrO_4 , finding values that are 34 and 156 meV/atom above the convex hull. Then then state: “We conjecture that our calculated ΔE^D data for Ca_2IrO_4 could serve as an approximate working range for the experimentalists.” and later claim that: “The remaining eight compounds have $\Delta E^D + 82$ meV/atom and fall within the bounds determined for the metastable Ca_2IrO_4 (discussed earlier). Therefore, we are cautiously optimistic that these eight compounds could also be synthesized under appropriate synthesis conditions, despite their positive ΔE^D values.”

Their conjecture is wrong. The calculated values for Ca_2IrO_4 do not form a range between which the decomposition energies of synthesizable compounds fall. Instead, the two different values for very similar structures indicate that one of the structures (the one with an energy that is 122 meV/atom higher) is not the experimentally observed state. An alternative explanation would be that the margin of error for their

calculated stabilities is on the order of 120 meV/atom, but that would invalidate the rest of the claims made in the manuscript.

Based on their calculations, a reasonable range for synthesizable structures would be those with decomposition energies of less than 34 meV/atom, not those with decomposition energies between 34 and 156 meV/atom. This leaves NaGdRuO₄ and NaYRuO₄ as predicted NCS structures that have a reasonable potential to be synthesizable. I suggest the authors revise their manuscript to more realistically present their results.

Response: We appreciate the comments. We have now revised our discussion for Ca₂IrO₄. We have deleted the following sentences on Page 9: “We conjecture that our calculated ΔE^D data for Ca₂IrO₄ could serve as an approximate working range for the experimentalists.” and “... and fall within the bounds determined for the metastable Ca₂IrO₄ (discussed earlier). Therefore, we are cautiously optimistic that these eight compounds could also be synthesized under appropriate synthesis conditions, despite their positive ΔE^D values.”

Referee notes that “The calculated values for Ca₂IrO₄ do not form a range between which the decomposition energies of synthesizable compounds fall. Instead, the two different values for very similar structures indicate that one of the structures (the one with an energy that is 122 meV/atom higher) is not the experimentally observed state.” We would like to clarify this point, because the two crystal structures for the metastable Ca₂IrO₄ are not quite similar. In Figure 1 (see Page 3 of this Response Letter), we show the crystal structures for Ca₂IrO₄ in the DFT optimized *I4/mmm* (high-symmetry) and *Pbca* (ground state) space groups. There are two important things to note here: (i) In the space group *I4/mmm*, the IrO₆ octahedron shows no tilting, whereas in *Pbca* it shows significant tilting. (ii) There is also cell doubling, where the in-plane lattice constants ($a = b$) in *I4/mmm* transforms to $\sqrt{2}a \neq \sqrt{2}b$ in *Pbca* structure. These distortions make the *Pbca* structure -121 meV/atom lower in energy relative to that of the *I4/mmm* structure.

Figure 1: DFT optimized crystal structures for Ca_2IrO_4 in (a) high symmetry $I4/mmm$ and (b) ground state $Pbca$ space groups.

REVIEWERS' COMMENTS:

Reviewer #3 (Remarks to the Author):

The authors have addressed my concerns, and I believe this manuscript can be published in its current form.

November 28, 2016

Letter to the Referees

Note: Our response to referees' comments are given in purple text color.

Report of the Third Referee – NCOMMS-16-01058C

Response to Referee #3

Comment: The authors have addressed my concerns, and I believe this manuscript can be published in its current form.

Response: We thank the reviewer for his/her comments and recommending our manuscript for publication.